

# Analysis of the simulated feedbacks on large-scale ice sheets from ice-sheet climate interactions

Zhiang Xie[1,2] and Dietmar Dommenget[2,3]

[1] Department of Earth and Space Sciences, Southern University of Science and Technology, Shenzhen, 518055, China
[2] Monash University, School of Earth, Atmosphere and Environment, Clayton, Victoria 3800, Australia
[3] ARC Centre of Excellence for Climate Extremes, Australia

*Correspondence to*: Zhiang Xie (xieza@sustech.edu.cn)

**Abstract.** In study presented here we focus on the large climate-ice sheet feedbacks on global scales on time scales of 100,000yrs. We conducted a series of idealised sensitivity experiments under $CO_2$ and solar radiation reduction scenarios with
the Globally Resolved Energy Balance - Ice Sheet Model v1.0 (GREB-ISM v1.0), to study the characteristics of five climate-ice sheet feedbacks, including albedo, snowfall, ice latent heat, topography and sea level feedbacks. We analysed the relative importance of each of these feedbacks on the ice sheet growth and on the climate system (surface temperature). The results indicate that the inclusion of ice sheets will delay the response to the external forcing and facilitate the climate cooling in the high latitude and altitude areas in the Northern Hemisphere, but also causes a small amount of warming elsewhere, due to the
blocking of atmospheric heat transport. As for individual feedbacks, the albedo feedback is the most dominant positive feedback in favour of ice sheet build-up and cooler climates, whereas snowfall feedback is the greatest negative feedback that reduces the growth of ice sheets. The large ice latent heat required to melt ice allows to maintain ice sheets from one cold seasons to the next and therefore provides a positive feedback for ice sheet growth. The ice sheets impact on the topography is also a positive feedback but with smaller impact than the albedo feedback. The sea level change influences ice sheets by
shifting their location, in particular allowing ice sheets growth in the Arctic Ocean, while reducing it over central north Asia.

## 1 Introduction

Ice sheets comprise an essential part of the Earth system that impact the climate on longer time scales and has shown substantial changes in the historical paleoclimate record (Kageyama et al., 2018). The ice sheets interact with other components of the climate system via different feedback mechanisms, which control the impact of the ice sheets on the climate system and in
turn also control the growth of the ice sheets. In this study we will evaluate five of the most important ice-sheet-climate feedbacks within climate model simulations. These feedbacks are: the ice-albedo, snowfall, ice latent heat, topography and sea level feedback.

The formation and retreat of ice sheets significantly alters the surface albedo. Snow/ice sheet surface albedo can reach values larger than 0.45 (Ryan et al., 2019), which is much higher than the global mean surface albedo (0.1). As a result, adjacent
surface temperature drops due to decreased absorption of shortwave radiation, which in turn leads to more snow-covered regions, defining a positive feedback loop (Gardner and Sharp, 2010). This leads to a tight relation between the surface temperature change and the albedo changes (Gardner and Sharp, 2010; Zeitz et al., 2021). The albedo feedback is usually



regarded as a most essential process influencing the climate change in ice age cycles (Budyko, 1969; Felzer et al., 1996; Fyke et al., 2018; Petit et al., 1999).

The most important source of ice sheet mass growth is snowfall accumulation, which is directly related to precipitation rates. Cooling of the surface temperatures that will allow to grow ice sheets will in general also reduce precipitation due to the reduced water vapour that colder atmospheres can hold. This creates a negative feedback for the ice sheets, as it will reduce the ice growth the colder the surface of the ice sheets becomes. The snowfall feedback is also frequently tied with the topography feedback, as decrease in precipitation owing to surface temperature drop, are also caused by the raised surface

height of ice sheets (Fyke et al., 2018; Medley and Thomas, 2019). Precipitation intensity is often also linked to mountain slopes, as steep topographical changes typically result in heavy precipitation (Fyke et al., 2018; Hakuba et al., 2012). Further, changes in the atmospheric circulation owing to ice sheet topography are also thought to be related to changes in precipitation during the ice age cycle (Löfverström and Liakka, 2016).

The ice latent heat needed to melt ice into water has a substantial impact on the surface energy balance (Hock, 2005). The

surface energy balance is impacted by the change in climatic state, which in turn affects the ice melting process (Ebrahimi and Marshall, 2016; Patel et al., 2021). In particular, the ice latent heat, as a medium between ice thickness and surface energy balance, is able to aggregate the signal from various seasons and thus exhibit a seasonal inertia effect. For instance, if there is additional snowfall accumulation in the cold season, the latent heat required to melt the ice during the warm season will delay the surface warming, shortening the warm season (Marshall and Miller, 2020). This seasonal inertial effect of the ice latent

heat is essentially a positive feedback that will allow ice sheets to grow faster by bridging the warm season.

Ice sheets further affect the climate system through modifying the surface topography, the so-called topography feedback. The topography feedback is a nonlinear one because it influences the snowfall rate and surface temperature simultaneously and in a non-linear way (Edwards et al., 2014; Hakuba et al., 2012). As an ice sheet grows, the elevated surface height lowers the surface temperature due to the lapse rate in the troposphere (Abe-Ouchi et al., 2007; Fyke et al., 2018). Meanwhile, the low

surface temperature and short air column due to surface lifting reduces the snowfall rate (Kapsch et al., 2021). Of course, the large topography created by massive ice sheets also has a considerable influence on the general atmospheric circulation (Felzer et al., 1996; Hakuba et al., 2012; Herrington and Poulsen, 2012). In paleoclimate simulation, those processes are believed to be connected with the temperature and precipitation change (Abe-Ouchi et al., 2007; Hakuba et al., 2012), deglaciation (Abe-Ouchi et al., 2013; Kapsch et al., 2021) and hysteresis of equilibrium states (Abe-Ouchi et al., 2013).

The total ice volume on Earth during the paleoclimate is intimately tied to global sea level change, which in turn affects climate via changing land-sea distribution and the topographic heights of land. Owing to absence of global coupled ice sheet model, the sea level is usually taken as external forcing in paleoclimate simulation (Pollard and DeConto, 2009; Tigchelaar et al., 2019). Actually, most of sea level change during the late Quaternary can be attributed to the Northern Hemisphere ice sheets change (Bentley, 1999). In addition, the sea level feedback is usually considered as an important feedback for ice shelf around

Antarctica Ice Sheet because the sea level change potentially causes the Marine Ice Sheet Instability (Schoof, 2007). As a result, the sea level feedback discussion usually focused on how sea level change caused by the Northern Hemisphere Ice Sheet impact the Antarctica Ice Sheet (Gomez et al., 2020; Maris et al., 2015; Tigchelaar et al., 2019).



In addition to the five feedbacks outlined above, many other feedbacks are related to the climate-ice sheet interaction. Pollard et al. (2015) pointed out that Antarctic Ice Sheet potentially retreated because of hydrofracturing and ice cliff failure, caused by the ice shelf basal melting. And both Abe-Ouchi et al. (2013) and Han et al. (2021) suggested the feedbacks associated with the solid Earth deformation takes effect during the ice sheet evolution. Moreover, Willeit and Ganopolski (2018) discussed the strong influence from surface albedo, and concluded that the existence of dust on surface snow potentially causes a large uncertainty in model simulation.

Due to high computational expense of long-term simulation, the relevant ice sheet-climate interaction studies commonly use a simple Earth system model, that is, by dynamically coupling the ice sheet model with a simplified climate model with low resolution and strong parameterization approach (Abe-Ouchi et al., 2013; Fyke et al., 2011; Ganopolski et al., 2010; Tigchelaar et al., 2019). This method is very close to the fully coupled simulation and thus shows much more details on the climate-ice sheet interaction. Most of those studies focused on the simulation of real ice age cycles driven by combination of greenhouse gas and orbital forcing. However, we also need some idealised sensitivity experiments to help us better understand the role of ice sheet-climate interaction since the importance and physical processes of individual feedbacks can be better explored in a simple setting.

As a result, in this study, we will conduct a series of idealised sensitivity experiments with GREB-ISM climate model (Xie et al., 2022), to explore ice sheet-climate interaction and associated physical processes. Our experiment design and related procedure will be explained in section 2. The findings of our sensitivity experiment will be summarised in section 3, along with a comparison of different feedbacks. The section 4 will go through each of the five feedback processes and discuss the mechanisms controlling each feedback. The final section, will summarize and discuss the study.

## 2 Model and method

### 2.1 Model

In this study, we use the Globally Resolved Energy Balance - Ice Sheet Model v1.0 ( GREB-ISM v1.0 , Xie et al., 2022), in which ice sheets and ice shelves are simulated on a global grid, fully interacting with the climate simulation of surface temperature, snowfall, albedo, land-sea mask, topography, and sea level. Thus, it is a fully coupled atmosphere, ocean, land and ice sheet model. The climate model is based on The Globally Resolved Energy Balance version 2.0 (GREB v2.0) model (Dommenget and Flöter, 2011; Stassen et al., 2019). The GREB consists of three physical layers, including atmosphere, surface and sub-surface ocean with horizontal resolution of 3.75° x 3.75° (96 x 48 points). In addition to the prognostic equations of the temperature of the three layers the GREB model is also simulating the atmospheric water vapor as a prognostic equation based on evaporation, precipitation, and atmospheric transport. Thus, precipitation is simulated in the GREB model as a result of the atmospheric state, following a simple diagnostic model by Stassen et al., (2019).

The simplicity of the model comes with the limitation that the dynamical mean state of the prognostic variables is relatively far away from the observed. This is addressed in the GREB-ISM by introducing flux corrections to the tendency equations of



the surface temperature and the atmospheric humidity to enforced a monthly climatology as observed for presented day external forcings (e.g. $CO_2$-concentraions and solar radiation).

Unlike atmospheric general circulation models, the GREB model does not simulate variations in the atmospheric circulation (weather), but has prescribed wind fields. Subsequently, the model has no variability other than the response to external forcings. As a consequence, the model state of a single year can be used as an estimate of the climate state for a given forcing

at the given time. No long-term time averaging is needed.

The ice sheet model part, simulates the prognostic equation for the ice thickness on the same horizontal grid as the GREB, including energy-based surface mass balance, ice temperature on 4 vertical layers, ice sheet dynamics and floating ice shelf processes (Xie et al., 2022). The ice sheet mass balance accumulates the simulated precipitation as snow rates, when both the surface and air temperatures are below or near the freezing point. The GREB-ISM v1.0 is able to simulate the 100,000 model

years within 24 hrs wall time on a standard personal computer, making long term simulations and a large set of sensitivity experiments feasible.

## 2.2 Simulated processes related to the ice sheet-climate interactions

The five feedbacks in our discussion interact strongly with one another through a number of physical mechanisms. For the

following discussions in the results section, it is important to explain some physical effects simulated in the GREB-ISM regarding the formation of an ice sheet and the interaction with the climate state. We have outlined several key mechanisms:

### *Ice-albedo effect*

The GREB-ISM albedo scheme is different from the original GREB. In the original GREB model (without ice sheet simulation)

surface albedo is diagnosed as function of surface temperature (Fig 1a). In the GREB-ISM the albedo is a function of ice thickness (Fig 1b).

### *Snowfall effect*

In the GREB-ISM model precipitation is diagnosed on the basis of atmospheric humidity, relative humidity, mean and standard

deviation of vertical atmospheric velocities. The latter two are external boundary conditions and the first two are simulated based on the state of the climate system. Thus, precipitation rates adjust to the climate state. Over land the precipitation rate are further flux corrected to match the observed precipitation rate to ensure realistic mass balances for continental ice sheets (Xie et al., 2022). This flux correction is designed in a way that the precipitation rates over land will still be able to change proportional to the zonal mean atmospheric humidity.


### *Elevation-surface temperature effect*

Higher elevated surfaces have colder temperatures, as they are in sensible heat exchange with an atmosphere that is colder than lower elevated surfaces, due to the adiabatic lapse rate in the troposphere. This effect is commonly included in ice sheet



modelling and plays critical role for ice sheet development (Albrecht et al., 2020; Fyke et al., 2011). Although the GREB-ISM
has only a single layer atmosphere, the surface temperature decreases with elevation due to sensible heat exchange with the
atmosphere, following a fixed dry adiabatic lapse rate (0.6K per 100m).

*Ice sheet atmospheric blocking*

Large continental ice sheets affect the atmospheric circulation, blocking air flow across the ice sheets. In early studies, air
transport was frequently associated with ice sheet growth via circulation changes caused by the topography change (Felzer et
al., 1996; Herrington and Poulsen, 2012). The GREB-ISM model simulates heat and moisture transport in the atmosphere but
assumes prescribed atmospheric mean wind conditions that do not respond to changes in the topography. However, the GREB-
ISM simulation of advection and diffusion of heat and moisture is scaled down for increased topography elevation, effectively
reducing atmospheric transport across high topography, thus responding to the formation of large ice sheets.


*Global sea level change*

The global volume of grounded ice sheets affects the global sea level in the GREB-ISM. This change in sea level affects the
elevation of land topography, the grounding lines of ice shelves and the land-sea mask for the climate. Figure 2 shows the
control bedrock, illustrating regions with shallow ocean points that will be most affected by sea level changes. An ocean point
that turns into a land point, is assumed to have a reduced soil moisture, and therefore has strongly reduced surface evaporation.
It also has no more interactions with the subsurface ocean layer. Both effects will affect the surface temperature heat balance
by reducing latent heat flux and subsurface ocean cooling.

**2.3 Design of sensitivity experiments**

The most prevalent external forcings in last million years are $CO_2$ concentration and solar insolation. Both have the ability to
alter the Earth's net radiation budget and thus induce global warming or cooling. As a result, the focus of experiments in this
study will be on understanding the climate-ice sheet feedback in the presence of $CO_2$ and solar insolation forcing.

A control simulation and a scenario simulation are included in each of our experiments. We first perform a 30 kyr control
simulation with current $CO_2$ concentrations and solar radiation, then a 100 kyr scenario simulation. We have two scenarios: a
$CO_2$ reduction scenario and a solar radiation reduction scenario. We designed the forcings of the FULL experiments to allow
the growth of large-scale Northern Hemispheric continental ice sheets. Therefore, we have tested several possibilities and
finally select the following settings: for the $CO_2$ reduction scenario, the $CO_2$ concentration drops from 340 ppm in the control
to 40 ppm in the scenario and the solar radiation remains as in the control run. Similarly, in the solar radiation reduction
experiment, solar radiation is reduced to 95% of control run, but $CO_2$ remains at the same level as in the control simulation.
To evaluate the sensitivity of the climate system to different processes, we conducted additional sensitivity experiments, in
which elements of the climate system are turned "off", following the approach in Dommenget and Floeter (2011). We created



five process switches in the GREB-ISM to examine the effect of each climate-ice sheet feedback, including albedo, ice latent heat, topography, snowfall, and sea level feedback (Table 1):

ALBD (albedo feedback) switch: The surface albedo of a snow/ice-covered points is higher than those of ice-free points. As a result, the surface albedo is a function of ice thickness in the GREB-ISM. If we turn off the ALBD switch, the surface albedo will be unaffected by changes in the ice thickness and will remain the same as in the control run (including its seasonal variations).

TOPO switch (topography feedback): In the GREB-ISM, surface elevation depends on a constant bedrock elevation and the simulated ice thickness. Surface elevation changes affect the sensible heat, precipitation, and atmospheric transport. When the TOPO switch is turned off, the model's surface elevation remains unchanged from the control run.

SNOW switch (snowfall feedback): Precipitation turns to snow in the GREB-ISM when both the surface and air temperatures
are below or near the freezing point. As a result, changes in precipitation during the snowy season will be directly reflected in snowfall rate. If we turn off the SNOW switch, however, the snowfall rate will remain the same as in the control run (including its seasonal variations), regardless of local precipitation changes.

HEAT switch (ice latent heat feedback): If the surface is covered with snow or ice during the melting season, a large amount
of surface heat flux will be used as ice latent heat to melt the surface snow or ice, rather than warming the surface temperature directly. In the melting season, this has a significant impact on the surface energy balance. If we turn off the HEAT switch in the GREB-ISM, the ice thickness change will still be calculated based on the energy balance but the ice latent heat will be set to zero in the temperature tendency equation.

SLV switch (sea level feedback): The GREB-ISM, as a global climate system model, simulates sea level based on the total grounded ice sheet volume. This will change bedrock elevation, affecting land-sea distribution and surface height, influencing climate and ice sheet simulations. The sea level will remain on the control run value if the SLV switch is turned off, not allowing any global sea level change.

Based on these process switches, we design the following five experiments: NALBD, NHEAT, NTOPO, NSNOW, and NSLV. Each N-X experiment represents the simulation with the GREB-ISM to illuminate the climate response without a specific feedback by turning off X but leaving all other switches on. To assure that all experiments are run for similar control climates, the flux corrections of the GREB model are recalculated for each of these experiments. Therefore, the global surface temperature in the control runs of all experiments are nearly identical, except for areas around Greenland and Antarctica in
NTOPO due to the decoupling of ice sheet and topography. For each experiment we conducted the same two 100 kyr scenario simulations with forcings in $CO_2$ concentrations and solar radiation.



In addition, we further conducted simulations with the GREB model not including the simulations of ice sheets (NOISM experiment). This experiment uses the GREB 1.0 (Stassen et al., 2019). In this experiment surface albedo is a function of surface temperature instead of ice thickness, and topography and sea level are fixed.

We define the temperature and ice thickness *response* to external forcing as the difference between the scenario equilibrium minus the control run. The overall effect of a feedback or process is evaluated by using the response difference between the FULL experiment minus the experiment with a specific feedback switch turned off ([FULL]-[EXP]). A positive feedback would show a difference with the same sign as the response in the FULL experiment and vice versa for a negative feedback. Similar to a radiative feedback framework (Goosse et al., 2018), a framework to evaluate the feedback strength for ice sheet

effect is used in our discussion. Assuming the feedback processes follow a linear model:

$$\frac{d\Phi}{dt} = c \cdot \Phi + Q \qquad (1)$$

where $\Phi$ is the variable we are interested, $Q$ is the external forcing and $c$ is the feedback strength. Given an external forcing

change $\Delta Q$ and equilibrium state change $\Delta\phi$, the feedback strength can be evaluated by

$$c \approx \frac{-\Delta Q}{\Delta \Phi} \qquad (2)$$

For instance, the albedo feedback strength $c_{ALBD} = c_{FULL} - c_{NALBD}$ can be evaluated by:


$$c_{ALBD} = c_{FULL} \cdot \frac{\Delta\Phi_{NALBD} - \Delta\Phi_{FULL}}{\Delta\Phi_{NALBD}} \qquad (3)$$

$c_{FULL}$ is directly evaluated by $\frac{1}{\Delta\Phi}$ for surface temperature (negative forcing) and $-\frac{1}{\Delta\Phi}$ for ice thickness (positive forcing), since the external forcing ($\Delta Q$) is the same for all sensitivity experiments.


## 3 Climate and ice sheet feedbacks

We start the results section with the analysis of the sensitivity simulations. First, we evaluate the impact of including an ice sheet simulation in the coupled model in section 3.1 and then we focus on the 5 different feedbacks and how they impact the ice sheets growth and the climate system in section 3.2. Detailed analysis of the feedbacks will be continued in section 4.

## 3.1 Ice sheet coupling to the climate

In order to examine the overall impact of the coupling between the ice sheet and the climate, we compare the coupled GREB-ISM (FULL) with the GREB model without the ISM (NOISM). Thus, ice sheets do not form in the NOISM simulations and



albedo is coupled to the surface temperature, but not to the ice thickness, as in the GREB v2.0 from Stassen et al. (2019). Figure 3 shows the global maps of the response in the surface temperature and changes in the topography, and Figure 4 shows the zonal mean changes of these variables.

Both model simulations show globally cooler surface temperatures with a global mean of about -7°C in response to both forcings (Fig. 4). The GREB-ISM (FULL) simulation shows stronger cooling over Northern Hemispheric land and in the Arctic than original GREB (NOISM) simulation. This stronger cooling in the GREB-ISM is largely a result of the increase topography due to the formation of large ice sheets (compare changes in Fig. 3). In the FULL simulations large ice sheets form in the high latitudes of at North America, North Eurasia, the Arctic and the Tibetan Plateau. Related to the large build-up of continental ice sheets is a reduction in the global sea level by 120 and 141 metres, for the $CO_2$ and solar radiation reduction experiments, respectively. This global sea level change leads to a global topography lifting of all land masses (Figs. 3g,h and 4b,d).

Interestingly, the FULL simulations also show somewhat weaker cooling than the NOISM simulation over Northern Hemispheric oceanic and most Southern Hemispheric regions. A series of shoreline grid points stick out with strong positive anomalies (Fig. 3e and f). These are gird points, which in the FULL experiments change from ocean to land points, due to the sea level reduction.

In addition to these equilibrium differences between the FULL and NOISM simulation we can note differences in the transient behaviour of the two simulations, in particular over the North Hemisphere high latitudes, see Fig. 5. While the climate reaches equilibrium with a few decades in the NOISM simulation, the FULL simulation requires more than 25 kyrs to get into equilibrium. This is due to the slow adjustment process that leads to the build-up of large continental ice sheets.

## 3.2 Sensitivity to different feedback processes

We now focus on how each of the five feedback processes influences the ice sheet formation and the climate. We therefore compare the FULL experiment with the one-switch-off experiments in the two forcing scenarios. In the FULL experiment (Fig 6a and b), the ice sheet response mainly concentrates on the Northern Hemisphere, where a vast land area is covered by ice sheets. The ice thickness anomalies are mainly located around 60°N. However, the climate-ice sheet feedbacks can either enhance (through a positive feedback) or reduce (via a negative feedback) this ice growth response to the external forcing. By evaluating the response difference of FULL experiment minus the one-switch-off experiments, the characteristic of each feedback can be quantified (Fig. 6c-l). Positive (negative) values in the ice sheet response difference indicate a positive (negative) feedback.

The largest impact by far is from the positive ice-albedo feedback. Without the ice-albedo feedback the ice sheet growth is essentially zero, as can be seen the zonal mean of ice sheet response in Fig. 7. The absence of the ice-albedo feedback eliminates any significant ice sheet response. The ice latent heat feedback is also positive, but much weaker than the ice-albedo feedback. The regional pattern of the response difference is similar to the overall response in the FULL experiment, indicating that this feedback is mostly an amplifying feedback, not altering the regional characteristics.



The snowfall feedback is the only significant negative feedback, which is of similar strength to the ice latent heat feedback, but with opposite sign. The snowfall feedback has a tendency to shift the ice sheet further north, which may partially be related to the larger land masses to the south of the ice sheets that form in the NPREP experiment. The snowfall feedback is also the only feedback, which has a significant influence on the Antarctic ice sheets.

The topography feedback is weaker than the above-mentioned feedbacks, but it is mostly a positive feedback. The sea level feedback in turn is more complex, as it tends to shift ice sheets in location rather than just amplifying or damping ice sheets growth. The sea level reduction due to the global growth of grounded ice sheets turns some of the shallow ice shelves in the Arctic Ocean to land points, which allows the growth of large ice sheets. This shifts the ice sheets from Asia towards the Arctic Ocean. Here it is interesting to note that the ice sheet growth is not only amplified over the Arctic Ocean, but also damped

over northern central Asia. This suggests that the build-up of the Arctic ice sheets does hinder the formation of a northern central Asian ice sheet.

Figures 8 and 9 show the response of the surface temperature in the FULL experiment and the response difference of the sensitivity experiments in reference to the FULL experiment. Starting with the albedo feedback, we can first of all note that the surface temperature response difference (Fig. 8c,d) is very similar to the FULL experiment response (Fig. 8a,b), indicating

that most of the regional differences in the surface temperature response results from the ice-albedo feedback. This is also illustrated in Fig. 9a,c where we can see that the NALBD experiment response is fairly similar on all latitudes, in contrast to the strong variations seen in the FULL experiments. The ice-albedo feedback is not only affecting the ice-covered regions, but does affect all latitudes (Fig. 9b), due to the atmospheric interactions (e.g., transport of heat and moisture). In contrast to this global positive feedback, there are coastal points in which the ice albedo feedback is negative (including the ice albedo

feedback leads to less cooling in response to the forcings; Fig. 8c,d). This results from the global sea level change, which will be discussed in more detail in the next section.

The snowfall feedback is a strong negative feedback over the regions with ice sheet growth, which directly results from the cooling effect of the surface elevation lifting by the ice sheets. Interestingly, lower latitudes see a positive feedback from the precipitation, similar in strength to the ice-albedo feedback (Fig.9 b,d). We will discuss the cause for this in more detail in the

next section. This aspect is similar for the topography feedback, but in general with opposite sign and somewhat weaker. The switch in sign of the feedback is also following closely the regions where the ice sheets grow (compare Figs. 6 and 8).

The ice latent heat feedback for the surface temperature is also following a similar pattern as the precipitation and topography feedbacks. While the direct response over the regions with ice sheet growth is similar in strength, it is weaker in the regions far away from ice covered zones. The same holds for the sea level feedback. Here we see positive feedbacks over ice covered

regions, following the ice sheet changes. In addition, we see negative feedbacks over ice-free regions similar to those in the NHEAT experiment.

In general, we find that all feedbacks have a direct feedback on the surface temperature over the regions with ice sheet growth, which directly results from the cooling effect of the surface elevation lifting by the ice sheets. In addition, all feedbacks have

an opposite sign feedback on the surface temperature over remote ice-free regions with varying strength. The exception is the



ice albedo feedback, which is positive globally. We will discuss the cause for these opposite sign remote feedbacks in more detail in the next section.

We can quantify the strength for each feedback based on Eq. (3) with $\Delta\Phi$ as the global mean ice sheet volume or surface temperature response. The results for each experiment and for both scenarios are shown in Figure 10. The overall feedback
strength the for FULL experiments is negative in both scenarios and both variables, as expected for stable global climate system. We can further note that the sum of all the one-switch-off experiments does not add up to the feedback strength of the FULL experiment. This is expected, since not all important feedback processes have been considered in these five experiments, including some essential negative feedbacks, such as the Plank radiative feedback, ice transport and ice sheet size feedback (Fyke et al., 2018).

The ice-albedo feedback is by far the most important feedback from the five feedbacks discussed for both variables. It is, however, more dominant for the ice sheet volume than for the surface temperatures. The ice latent heat feedback is the second strongest feedback for the ice volume, but is somewhat weaker for the surface temperature, where the snowfall feedback is about as strong.

The latter is an interesting finding, since the snowfall feedback has significant opposite sign feedbacks over ice-free regions,
which the ice latent heat feedback does not have as strongly. This should reduce the strength of the snowfall feedback as we only consider global mean values in Figure 10. This is likely affected by the non-linear aspects in the estimation of the feedback strength (Eq. 3). Since the snowfall feedback is the only significant negative feedback over ice sheet regions, the NSNOW experiment has by far the largest extent in continental ice sheets. This affects the linear approximation of Eq. (3).

The topography feedback is significant for the ice volume, but near zero for the surface temperature, as the feedback has
opposite signs for different regions. Similarly, the sea level feedback is overall fairly weak, which is also due to its counteracting effect by shifting the location of ice sheets.

Our results here agree with numerous earlier investigations, which established that albedo feedback has a dominating role in Northern Hemisphere climate change during paleoclimate (Abe-Ouchi et al., 2007; Budyko, 1969; Felzer et al., 1996). However, there are some differences between our study and the previous works. First, those early studies only evaluate the
effect from the ice-albedo and topography feedback and few of them discussed the ice latent heat, snowfall and sea level feedbacks. Second, the topography feedback in those early studies also included the atmospheric circulation changes, such as stationary wave patterns, which are absent in our study.

## 4 Analysis of the feedback processes

We now focus on the individual feedback processes in more detail, aiming at getting a more in depth understanding of how
theses feedbacks work and what is causing some of the unexpected characteristics.



## 4.1 Albedo feedback

High surface albedo is an essential feature of snow-covered area. As a result, adjacent surface temperature drops due to less shortwave radiation abortion. The process of albedo feedback is quite clear. The initial surface cooling caused by solar radiation and $CO_2$ reduction favors ice sheet formation, which in turn further increases the surface albedo in most of Northern
Hemisphere land by more than 5% (Fig 11a, c). The brighter ice surface leads to more than 10 Wm$^{-2}$ annual surface solar radiation reduction in the whole Northern Hemisphere land, facilitating the initial surface cooling from the external forcing (Fig 10b, d). The effect relatively strong in the Arctic, as the albedo changes are in summer, while it is not as strong in lower latitudes, as here the albedo changes are mostly in winter (not shown). In general, the ice-albedo feedback in our simulations is conceptually similar to what has been described in previous studies (Fyke et al., 2018; Willeit and Ganopolski, 2018).
The above physical process of the ice-albedo feedback does not quite explain why the FULL experiment has such a strongly amplified surface temperature response for large ice sheets (compare Figs. 7 and 9). This is related to the surface elevation lifting or the topography feedback, which will be discussed further below. It is worth noting that we did not systematically evaluate albedo scheme changes (e.g., changes in the albedo of ice). Willeit and Ganopolski (2018) pointed out that the Northern Hemisphere ice sheets over the last glacial cycle is very sensitive to the representation of snow albedo in the
modelling. This is an interesting subject that warrants further investigation.

## 4.2 Snowfall feedback

Snowfall rate is the primary source of ice mass in GREB-ISM. When the surface and air temperatures drop near or below the freezing point, precipitation turns to snow in the GREB-ISM. Most previous studies linked the snowfall feedback directly to topography changes (Fyke et al., 2018; Hakuba et al., 2012; Löfverström and Liakka, 2016; Medley and Thomas, 2019). In
the GREB-ISM, the precipitation is simulated based on the state of the atmosphere and is related to local or zonal mean atmospheric water vapor capacity (see section 2.2), so it is indirectly connected with surface temperature and also the topography.

The cooler climate in the scenario runs lead to reductions in the precipitation rates that affect the snowfall rates in different ways. Figure 12 shows the changes in the snowfall rates in the northern hemisphere for the winter and summer seasons in the
FULL experiment. We can see a clear reduction in snowfall rates in higher latitudes in winter and increases in snowfall rates in lower latitudes (Fig. 12a). The reduction at higher latitudes results from the cooler atmospheric temperatures that reduce the atmospheric water vapor concentrations, and therefore also reduce the precipitation and snowfall rates. At lower latitudes this is also the case, but at the same time the cooler temperatures in lower latitudes allow longer snowing seasons. This leads to increase in snowfall rates in winter (Fig. 12a). This effect dominates in summer also at higher latitudes (Fig. 12b).
The combined effect of decrease snowfall in winter and increased snowfall in summer at higher latitudes leads to an overall decrease in the annual mean snowfall rates (Fig. 13) reducing the ice sheets growth. The development of the large continental ice sheets further reduces the snowfall rates, due to the lifting of the surface elevation, which reduces the precipitation (Fig. 12c,d). In summary, the snowfall rates are decreasing during the ice sheet formations, due to the colder climate and the surface elevation lifting. This is a negative feedback for the ice sheet growth.



### 4.3 Ice latent heat feedback

The ice latent heat required to melt ice is substantial. To melt a 1 m ice column requires $3.03 \times 10^8$ J m$^{-2}$ energy, which is roughly equivalent to warm the land surface by 60°C. This additional heat required to change the surface temperature of ice covered regions does provide a substantial additional inertia to the climate system. It will in general delay the climate response to heating. In the context of the seasonal cycle it can help to overcome the warm summer season and allow the ice sheets to accumulated from one winter to the next winter.

We can illustrate this effect by considering a location in northern central Asia (90 °E, 70 °N), see Figure 14. In the FULL experiment, compared with NHEAT experiment, during the spring, a portion of the surface energy flux is consumed by ice melting (Fig 14c, f), resulting in a decrease of the surface temperature (Fig 14a, d). This cooling impact reduces the number of positive degree days in the following months (Fig 14a, d), which increase the ice-covered period (Fig 14b, e). As a result, more surface shortwave radiation is reflected during the warm season (Fig 14c, f), further reducing the temperature (Fig 14a, d). After multiple repetitions, it ends up with a permanent ice cover zone.

This seasonal effect of ice latent heat feedback works in regions with the short melting seasons, where the decreasing melting season length eventually causes irreversible ice accumulation (Fig. 15). First, we can note that the regions around the arctic have a short season of positive degree days (PDD) in the control climate (Fig. 15a). These are also the regions where the differences in the warm season temperatures and build up of ice sheets in the first decades are largest between the FULL and the NHEAT simulations (Fig. 15b-e). In the FULL simulation, including the ice latent heat feedback, the summer temperatures decrease faster and the ice sheet build up more strongly.

### 4.4 Topography feedback

In our sensitivity experiments, following a significant cooling caused by $CO_2$ and solar radiation decrease, large ice sheets start growing, eventually raising the surface elevation to thousands of metres (Fig 6). This causes the surface temperature to cool down as it is now exchanging a sensible heat flux with an ambient atmosphere that is much cooler due to the lapse rate in the troposphere (see section 2.2). Figure 16 shows how the surface temperature and the surface elevation change relate to each other. It clearly follows the fix dry adiabatic lapse rate used in the GREB model. Thus, most of the strong cooling over the large ice sheets is a result of the tropospheric lapse rate cooling. This feedback is widely discussed and also referred to as "elevation-surface mass balance feedback" (Abe-Ouchi et al., 2007; Edwards et al., 2014; Fyke et al., 2018), which is a positive feedback during deglaciation period (Abe-Ouchi et al., 2013; Kapsch et al., 2021).

The development of large continental ice sheets also affects the atmospheric circulation, blocking the flow of air across the newly formed mountain ranges. This blocking effect can be noted in the surface temperature response changes (Fig. 7i,j), where surface temperature poleward of the ice sheets is cooler in the FULL experiment than in the NTOPO experiment and warmer equatorward. It can also be seen in the other sensitivity experiments (Fig. 7), but with different strength or sign, depending on the nature of the feedback under consideration.



We conducted an additional sensitivity experiment in which we prescribed only the change in topography of [NPREP] – [FULL] without any other external forcing, see Fig. 17a. We run this experiment for 100 yrs to allow the atmosphere and oceans to get into equilibrium, but do not allow the ice sheets to grow, as this would lead to further topography changes. Here we can clearly see how the new mountain ranges block the air flow (heat and moisture) leading to cooler surface temperature poleward of the mountain ranges and warmer equatorward. Thus, the topography feedback has two aspects: it is a positive feedback for the ice thickness and surface temperature locally where the ice sheets form, and it is a negative feedback remotely, due to the blocking of the atmospheric air flow.

### 4.5 sea level feedback

In the GREB-ISM, the global sea level change affects the land-sea distribution and the global surface land elevation (see section 2.2). In the $CO_2$ and solar radiation reduction experiments, the sea level lowers by 120 and 141 metres, respectively for the FULL simulation. As a result of the global sea level drop, all ocean grid points with bedrock shallower than the global sea level change (Fig. 2) become land points (Fig 18). Further, the land surface elevation is lifted relative to the sea level.

Many shallow shelf seas in the Arctic ocean become land points in the FULL experiment (Fig 18), which allows the faster growth of ice sheets in these locations. The growths of ice sheets in these Arctic Ocean points blocks the air flow between the arctic ocean and the continental regions. This decreases the ice sheets grow on the continental regions, effectively shifting the ice sheets from continental regions to the Arctic Oceans (Figs. 6k,l and 7).

Coastal points in warmer regions (without ice cover) that turned from ocean to land points show a clear warming pattern (Fig. 8k,l). This, is due to changes in the latent heating and subsurface ocean heat exchange. We can illustrate this change by analysing the heat budget changes at a location at the Australian coastline in the tropical Pacific (140°E, 10°S), see Fig. 19. The transition from ocean to land at this location occurred between 3 and 4 kyr. This leads to a transition of the soil moisture to be much lower and the subsurface ocean heat change to vanish. As a result, the surface energy loss due to surface latent heat reduces by roughly 30 W m$^{-2}$ (Fig 19b, d) and the missing cooling of the subsurface ocean increase the heat budget by about 2 W m$^{-2}$. Sensible heat and net surface longwave radiation begin to compensate for the warming, until a new surface energy balance has been achieved (Fig 19b, d).

The global sea level change also lifts all land masses relative to the oceans. Given the fixed lapse rate in the atmosphere this leads to a cooling of all land point by about 0.6 °C. We conducted an additional sensitivity experiment in which we prescribed only a 140 m reduction in global sea level without any other external forcing, see Fig. 17c,d. We run this experiment for 100 yrs to allow the atmosphere and oceans to get into equilibrium, but do not allow the ice sheets to grow, as this would lead to further sea level changes. The experiment illustrates how the global sea level reduction led to cooling of all land points by about 0.6 °C, which is further amplified at higher latitudes by the ice-albedo feedback. It also illustrated how the coastal points that turn from ocean to land are warming substantially.



## 5. Summary and discussion

In this study, we explore the climate-ice sheet interaction by a series of sensitivity experiments in $CO_2$ and solar radiation
reduction scenarios focussing on global-scale and 100,000 yrs time scales. The analysis is based on model simulations with
the GREB-ISM. The results of these sensitivity experiments allow us to explore the role ice sheets play in the global climate
system and how different feedbacks within the climate system affect the ice sheets growth and its impact on the global climate
system.

The response of the ice sheets to changes in the climate system or external forcings enhance the climate response at higher-
latitudes due to surface elevation and albedo change. The inertia of the ice sheets also slows down the response of the climate
system to time scales of about 1000 yrs rather than the 10-100 yrs time scales of the atmosphere-ocean system. Further, the
large topography changes created by ice sheets prevent heat and water vapor transport across the ice sheets, cooling the high
latitudes and warming the low latitudes. This leads to a warming effect for most regions not covered by ice.

The global impact of ice sheets is fairly limited in the GREB-ISM simulations, if the albedo feedback is not considered. While
locally the ice sheets have a substantial impact, due to the lifting of the surface elevation, but remotely the effects are minor.
Here it needs to be considered that the remote influence is largely a result of the ice sheets affecting the atmospheric circulation.
This can also be altered by affecting the ocean circulation. The GREB-ISM model does simulate some aspects of the changes
in the atmospheric heat transport, but not in the oceanic heat transport. Further studies with more realistic simulations of
changes in the atmospheric and oceanic circulation need to be conducted to better understand the global impact of ice sheets.

We further focused on different climate-ice sheet feedbacks in the experiments. The albedo feedback is the most significant
positive feedback for both ice thickness and surface temperature. Massive ice sheets in the North Hemisphere are difficult to
build without cooling from albedo feedback. Given the dominance of the albedo feedback, details of the ice albedo are like to
be important for understanding the global impact of ice sheets. This should include variations of the albedo for different types
of ice or the influence of other factors, such as dust, on the ice albedo. This was not considered in the GREB-ISM simulation,
but should be addressed in further studies.

The snowfall feedback suppresses the growth of the ice sheet and temperature cooling by decreasing snowfall during the cold
season. The decrease in snowfall is mostly caused by the colder winter climate, but is also amplified by elevation increases.
Depending on the latitudes this negative feedback can be compensated or even turned into a positive feedback by increased
snowing seasons in summer.

The ice latent heat required to melt ice sheets does control the response time of ice sheets. This can have a strong effect on ice
sheets growth, when considered for the seasonal variations. It allows ice sheets that build up in the cold seasons to exist through
the warm melting season and therefore allow the growth of permanent ice sheets. This seasonal effect can be considered a
positive feedback of ice sheets growth.

The topography increase by ice sheets is a positive feedback for the ice sheet growth and surface cooling. By lifting the surface
elevation the ambient air will become cooler due to the adiabatic lapse rate in the atmosphere allowing ice sheets to build
further. The mountain ranges that are created by large ice sheets also affect the atmospheric heat and moisture transport. This
will mostly lead to a small warming effect equatorward of the ice sheets and a further cooling effect poleward.

Finally, decreasing sea level due to the build-up of large continental ice sheets shifts the locations at which ice sheets can form. This is in particular the case in the Arctic Ocean where ice sheets start to from due to the lower sea level, which in turn block the ice sheet formation in central northern Asia. The global sea level change also affects all continental regions by lifting them relative to the sea level. This has a small cooling effect over land.

The above discussion of feedbacks is clearly somewhat idealised and does not include all important aspects. As already mentioned above a more detailed studied of how atmospheric and ocean circulation changes due to the build-up of ice sheets is important for a better understanding of the global impact of ice sheets. This was also recognized as important factors in previous studies (Felzer et al., 1996; Larour et al., 2012). We further also neglected additional feedbacks such as ice sheets transport or interaction with the carbon cycle that can have important on the global climate system. Finally, the simulation conducted here only considered the build-up of ice sheets, but did not consider a melting phase. Studying the later will also be important to fully understand the ice sheets feedbacks. The GREB-ISM model can address such problems, but may also need further development to address some more complex aspects.

## 6 Code availability

The GREB-ISM source code, the model input data as well as a simple user manual are available on Zenodo: https://zenodo.org/badge/latestdoi/372993505. The model license is Creative Commons Attribution 4.0 International.

## 7 Author contributions

Zhiang Xie designed and ran all experiments and finished the first draft. Dietmar Dommenget termed the initial concepts, helped design the experiments and revise the paper draft.

## 8 Competing interests

The authors declare that they have no conflict of interest.

## 9 Acknowledgements

This study was supported by the Australian Research Council (ARC) Centre of Excellence for Climate Extremes (Grant Number: CE170100023).

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




**Table 1 Climate – ice sheet feedback and process switches.**

| Feedback | ON | OFF |
|---|---|---|
| Albedo (ALBD) | Albedo change due to ice thickness | Albedo fixed as initial condition |
| Latent heat (HEAT) | Ice latent heat is negative when melting | Ice latent heat set as zero |
| Topography (TOPO) | Topography change due to ice thickness, which is related to surface sensible heat, precipitation and heat transportation | Topography fixed as initial condition |
| Snowfall (SNOW) | Snowfall based on local precipitation | Snowfall fixed as initial condition |
| Sea level (SLV) | Sea level change due to global ice volume change | Sea level set as zero |






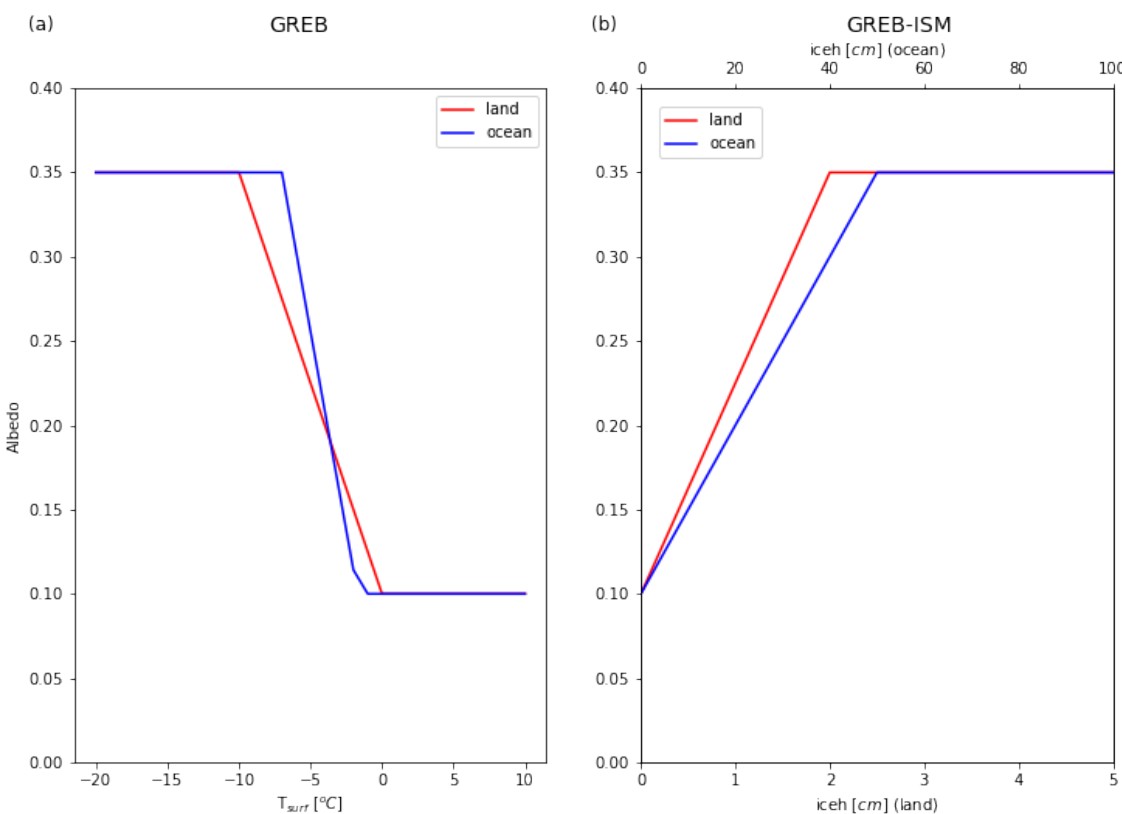

**Figure 1: The parameterization of the surface albedo in the original GREB (a)  and the GREB-ISM (b).**



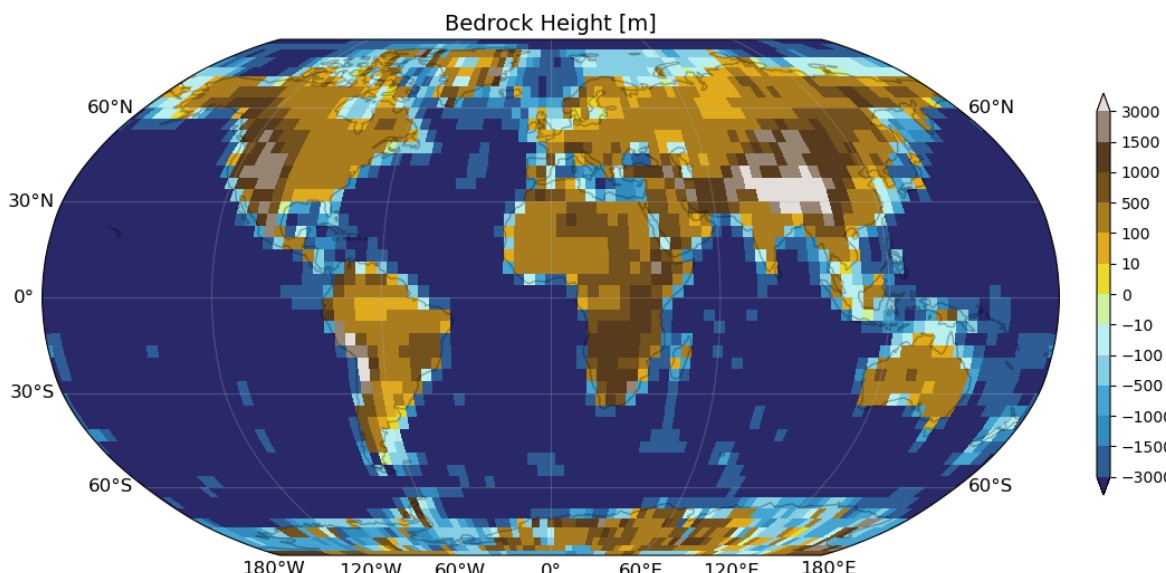

**Figure 2: Bed rock (unit: m) in the GREB-ISM.**






**Figure 3: Temperature response (unit: ºC) comparison between NOISM experiment (a, b), FULL experiment (c, d), their difference (e, f; FULL – NOISM, values are plotted on a logarithmic scale) and surface elevation change (g, h) in FULL experiment under $CO_2$ (left column) reduction and solar radiation reduction (right column) scenario. The temperature response is defined by using equilibrium ice thickness in scenario experiment (100 kyr) minus control experiment (30 kyr for FULL experiment, 50 yrs for NOISM experiment).**





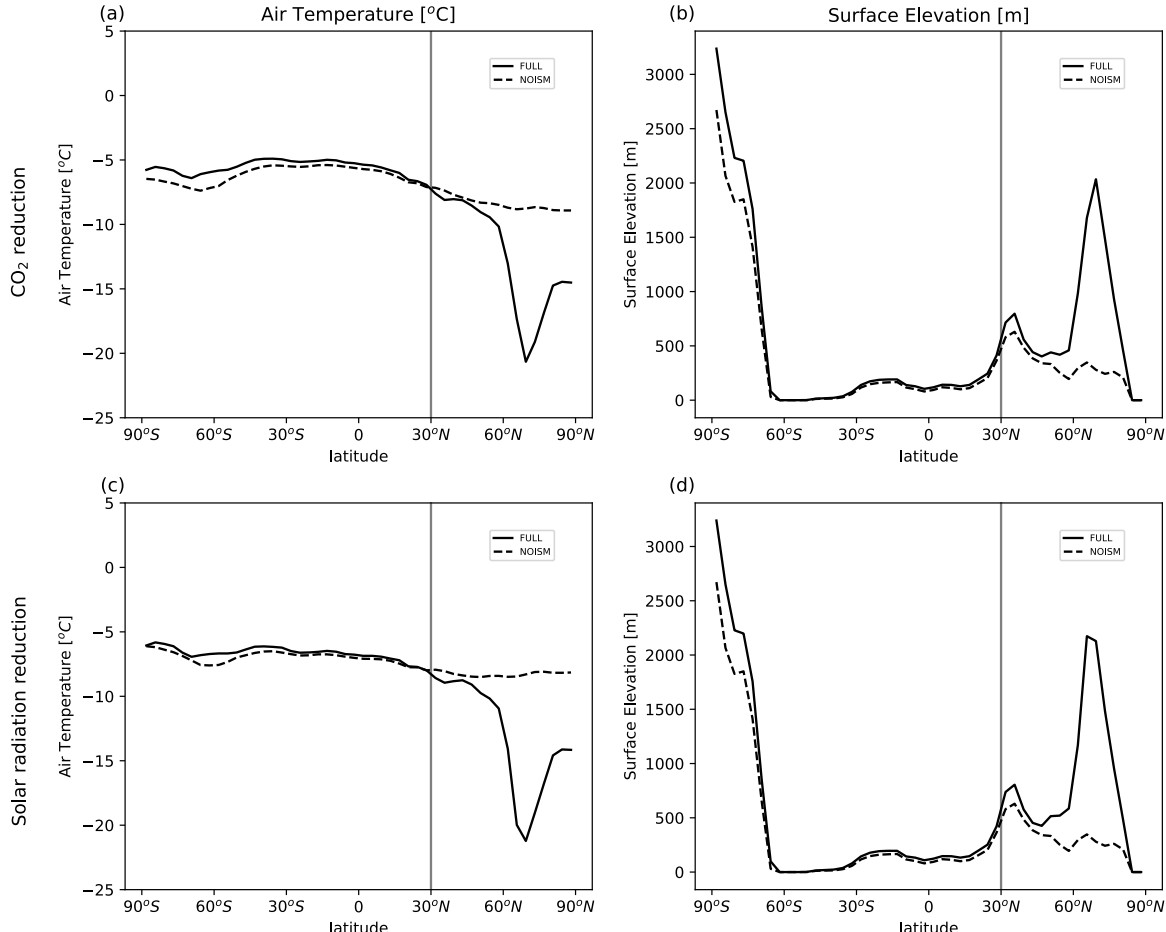

**Figure 4:** **Zonal mean surface temperature response (a, c; unit: ºC) and surface elevation (b, d) comparison between (dash line) NOISM experiment and (solid line) FULL under $CO_2$ reduction (upper row) and solar reduction (lower row) scenario. The temperature response is defined by using equilibrium ice thickness in scenario experiment (100 kyr) minus control experiment (30 kyr for FULL experiment, 50 yrs for NOISM experiment). The gray line represents 30ºN latitude.**



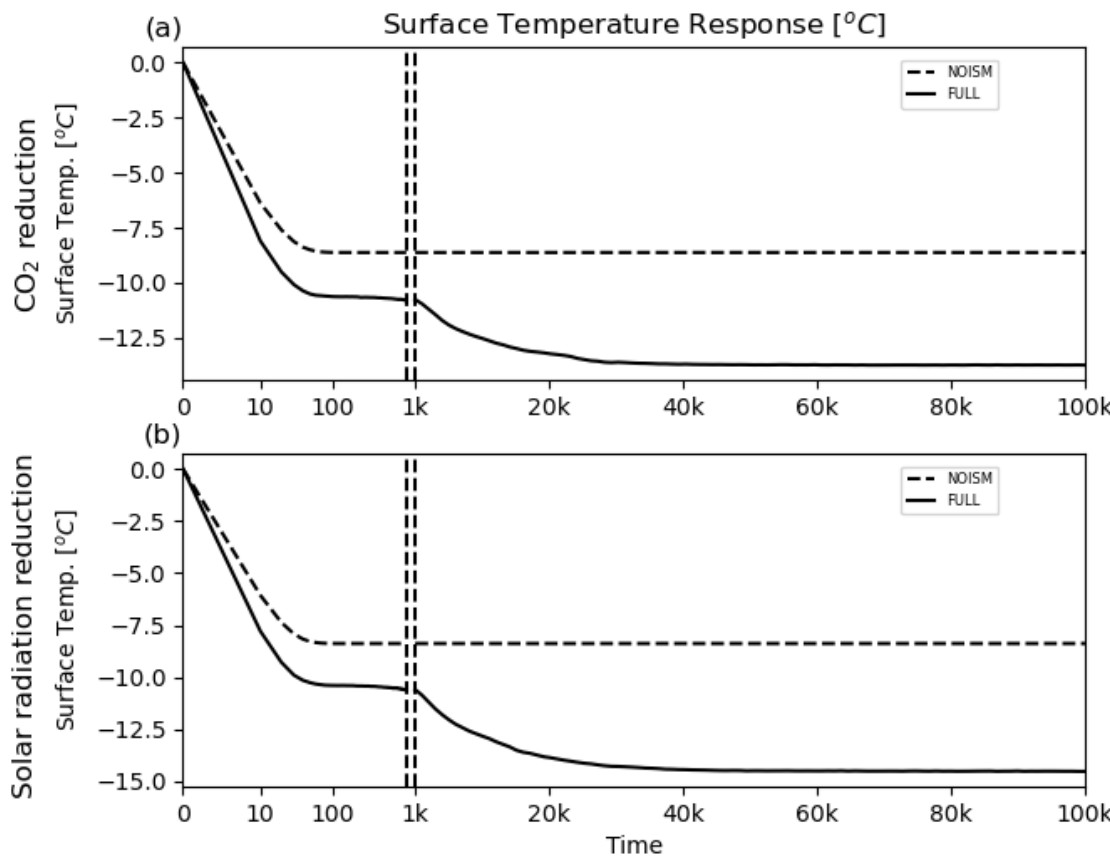

**Figure 5: The time evolution of North Hemisphere high latitude (50 ºN north) surface temperature response (solid for FULL experiment, dash for NOISM) under CO₂ reduction (a) and solar radiation reduction (b) scenario.**



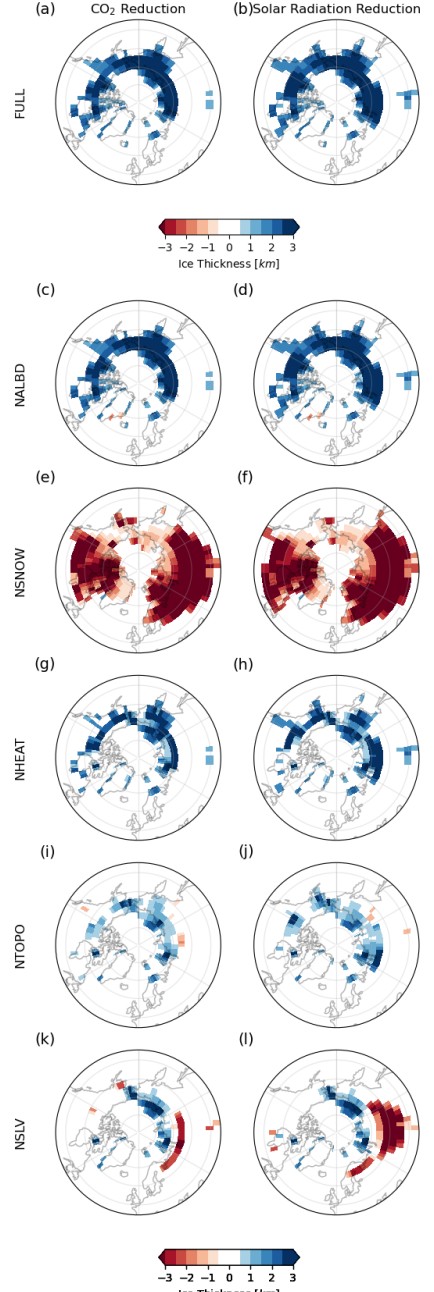

**Figure 6: Ice thickness response (a,b; unit: km) for FULL experiment and feedback effect (c-l) for individual feedback under $CO_2$ reduction (left column) and solar radiation reduction (right column) scenario. The ice thickness response is defined by using equilibrium ice thickness in scenario experiment (100 kyr) minus control experiment (30 kyr). The feedback effect is defined as the ice thickness response of the FULL experiment minus experiment without the specific feedback. "N-" represents experiment without specific feedback.**




**Figure 7: Zonal mean ice thickness (a,c; unit: km) in scenario run and zonal mean ice thickness [FULL] – [EXP] (b,d; unit: km) for individual feedback under CO₂ reduction (upper row) and solar radiation reduction (lower row) scenario.**




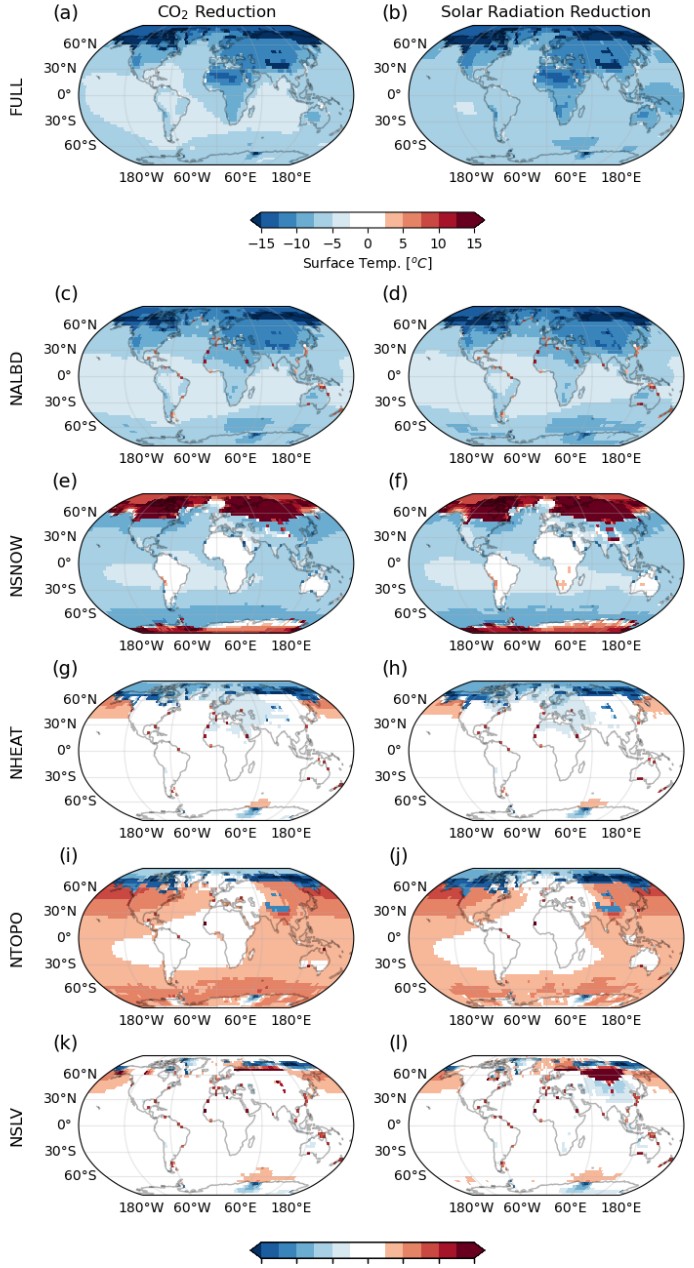

**Fig 8 Same as Fig 6 but for surface temperature (unit: ºC). Please note we use log scale colors in panels (c-l).**







**Fig 9 Same as Fig 7 but for surface temperature (unit: °C).**





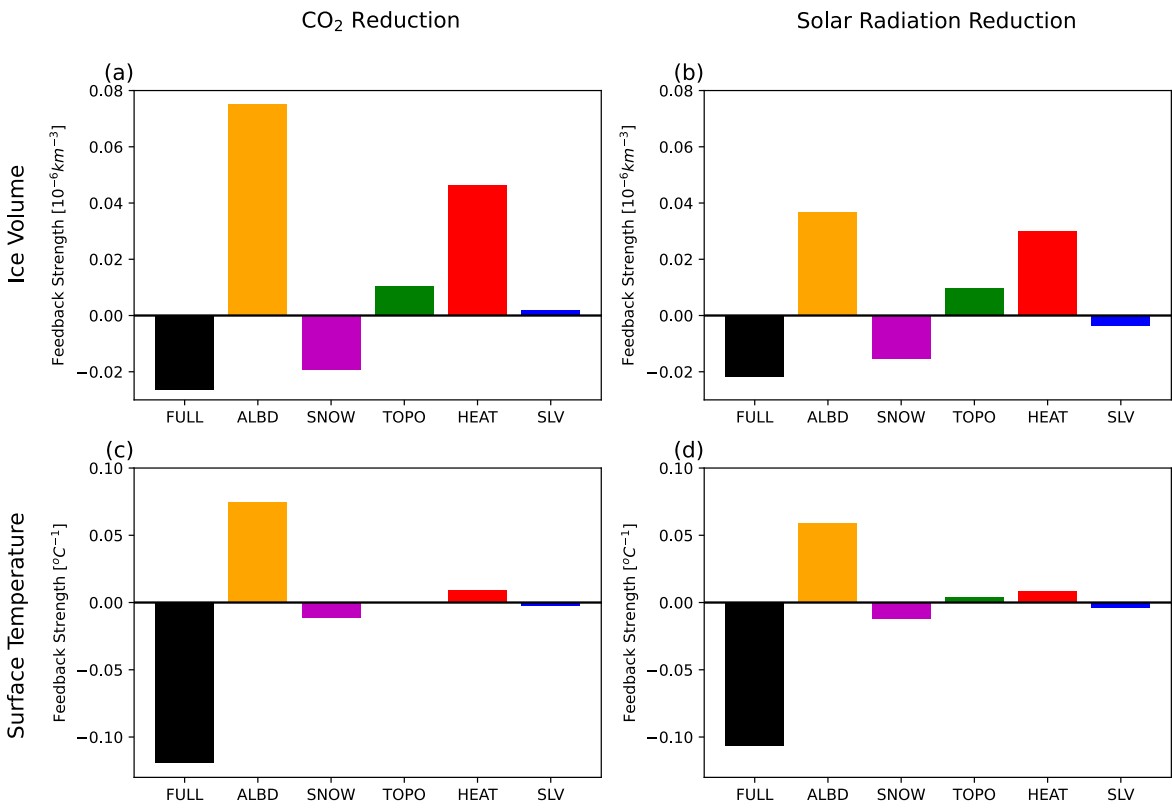

**Fig 10 Feedback strength of global total ice volume (a, b) and global mean surface temperature (c, d) in different experiments under CO₂ reduction (left column) and solar radiation reduction (right column) scenario. The feedback strength is defined by Eq. (3). The albedo feedback strength in total ice volume has been divided by 10 to do better comparison.**





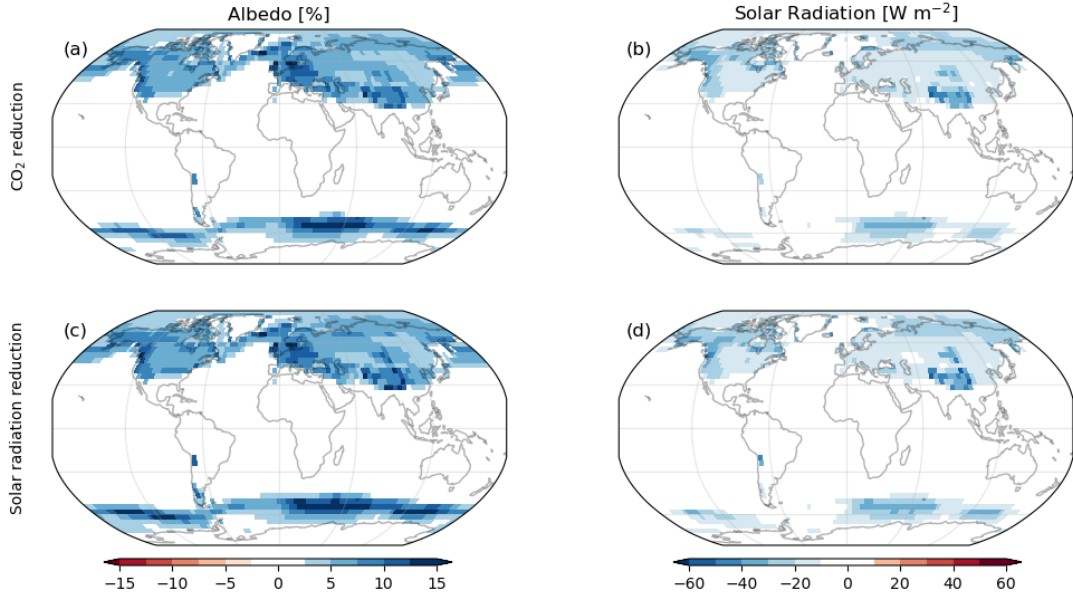

**Fig 11 Surface albedo (a, c; unit: %) and solar radiation (b, d; downward positive, unit: W m⁻²) between FULL and NALBD experiments ([FULL] − [NALBD]).**



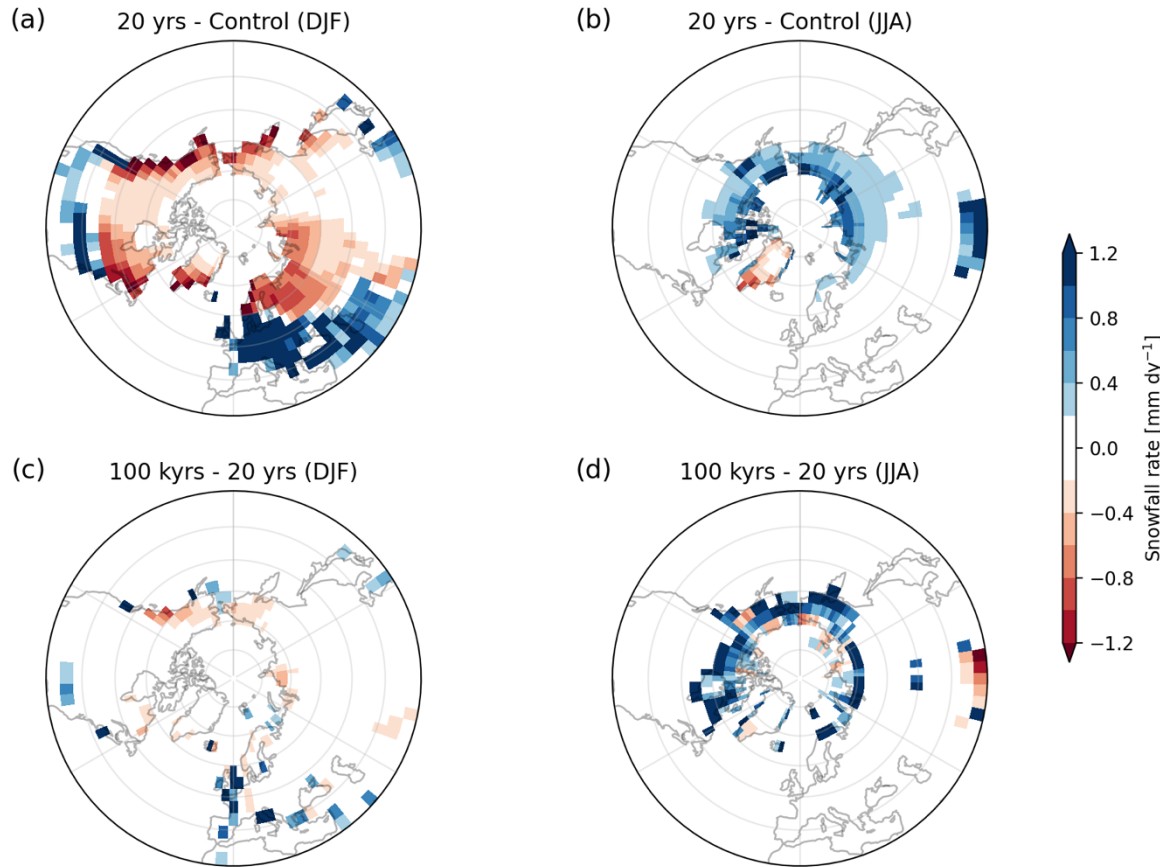

**Figure 12: Snowfall rate changes in winter (left column; DJF-means) and summer (right column; JJA-means) for the FULL experiment after 20 yrs (upper row) and changes within the FULL experiment after 100 kyrs (lower row). Values are in mm dy⁻¹.**



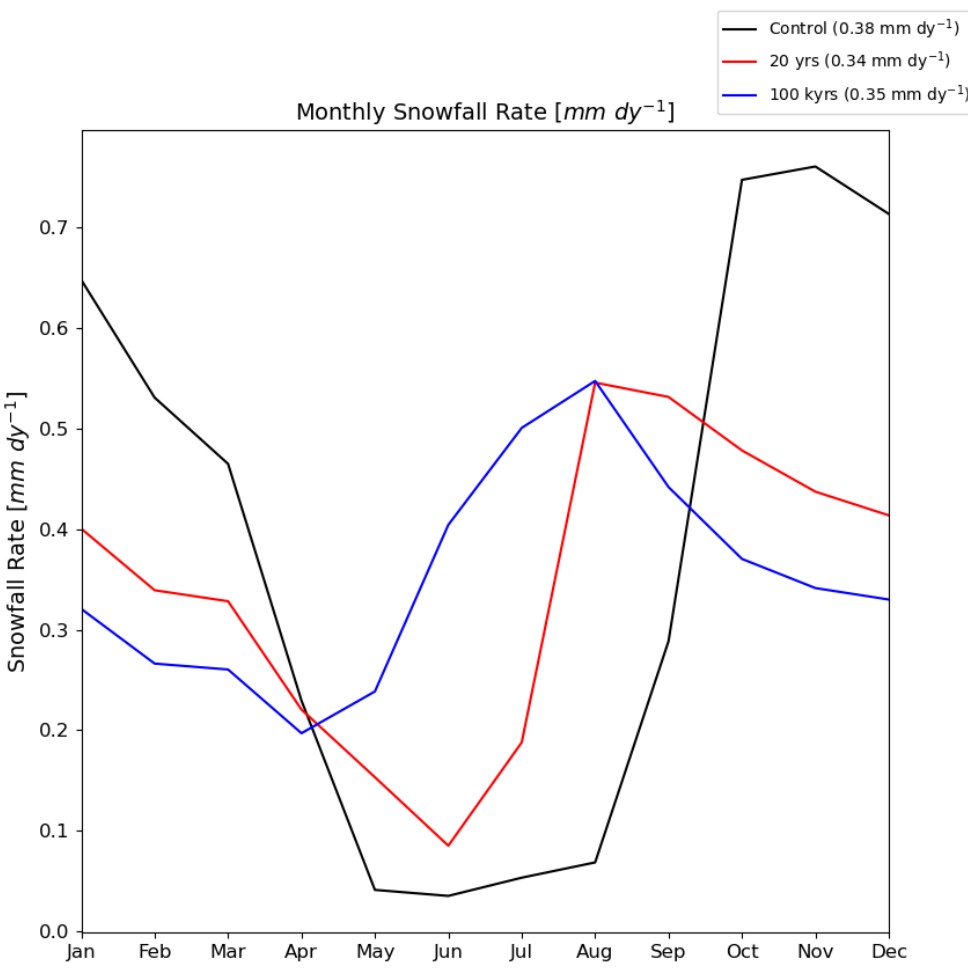


**Figure 13: Snowfall rates for land points 60 °N northwards in the control run (black), the FULL experiment after 20 yrs (red) and after 100 kyrs (blue) as function of calendar month. The numbers inside blankets represent the annual mean of snowfall rate.**



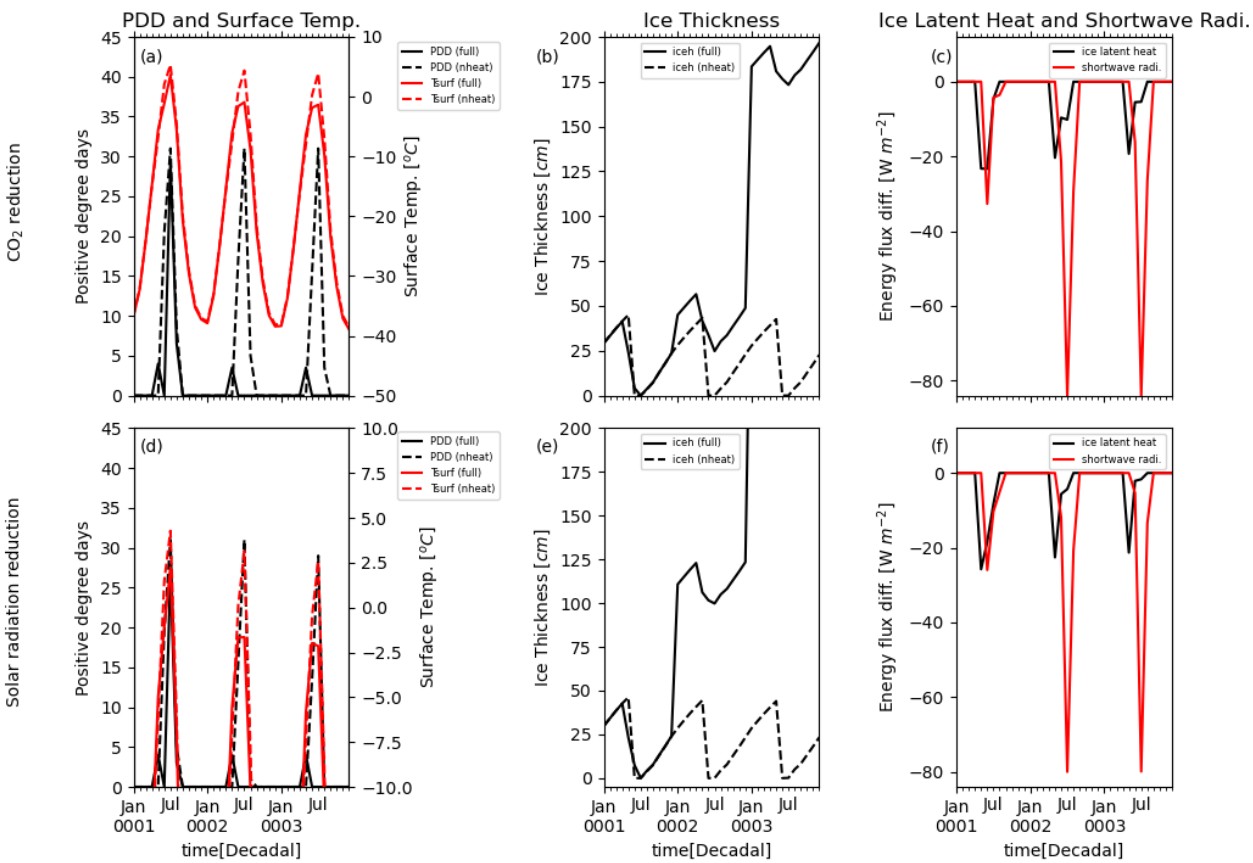

**Fig 14 Surface temperature and positive degree days (PDD) (a, d), as well as ice thickness (b, e) throughout the first three decades of the FULL and NHEAT experiments under CO₂ (upper row) and solar radiation reduction (lower row) scenarios. Ice latent heat and shortwave radiation difference (c, f) between FULL and NHEAT experiments in the first three decades under CO₂ (upper row) and solar radiation decrease (lower row) scenarios.**



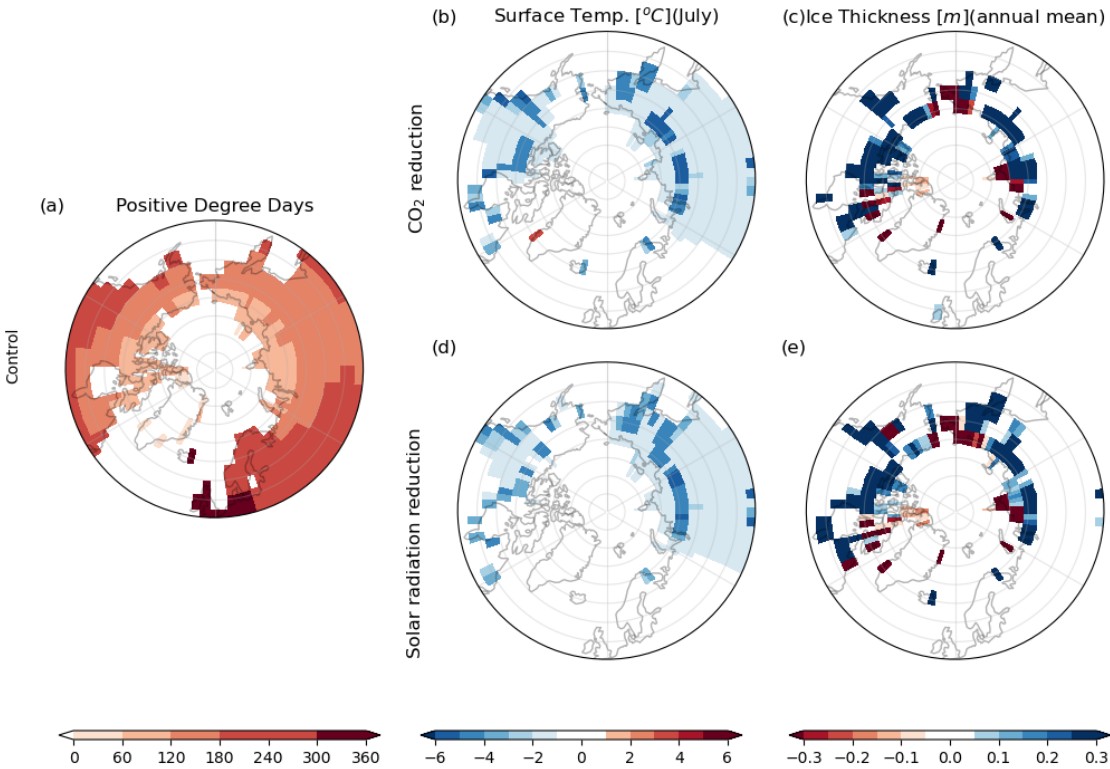

Fig 15 (a) Positive degree days (PDD) in FULL experiment scenario run after 20 yrs. July Surface temperature difference (b, e) and annual mean ice thickness difference (c, e) in 20 yrs between FULL and NHEAT experiments under $CO_2$ (upper row) and solar radiation reduction (lower row) scenarios.



Figure 16: Surface temperature vs surface elevation for all land points (blue for solar radiation reduction scenario and red for CO₂ reduction scenario) in FULL experiment in 60 ºN northwards in January.



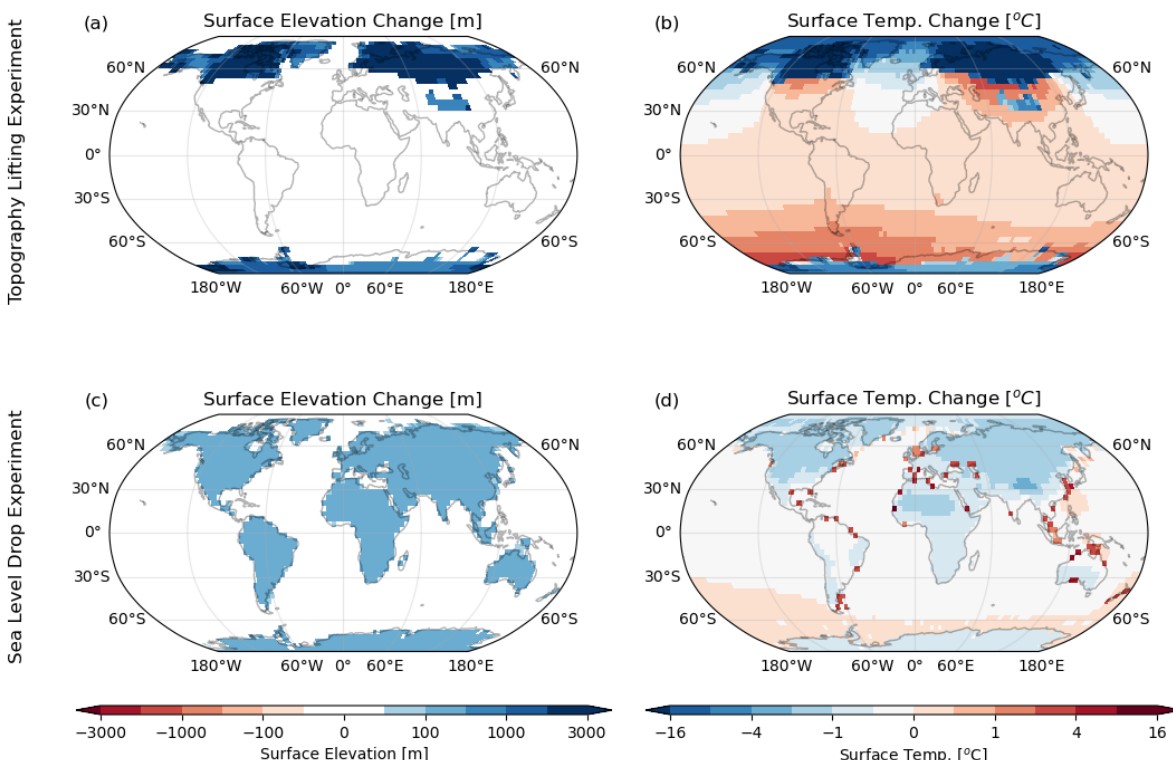

**Figure 17: Surface elevation change (a,c; unit: m) and surface temperature change (b,d; unit: ᵒC) in topography lifting experiment using NPREP – FULL surface elevation as forcing (upper row) and sea level drop experiment using 140 m sea level drop (lower row).**





**Fig 18 Land-sea mask change (blue represents from ocean to land grid) between FULL and NSLV experiment under CO₂ reduction (a) and solar radiation reduction (b) scenarios.**



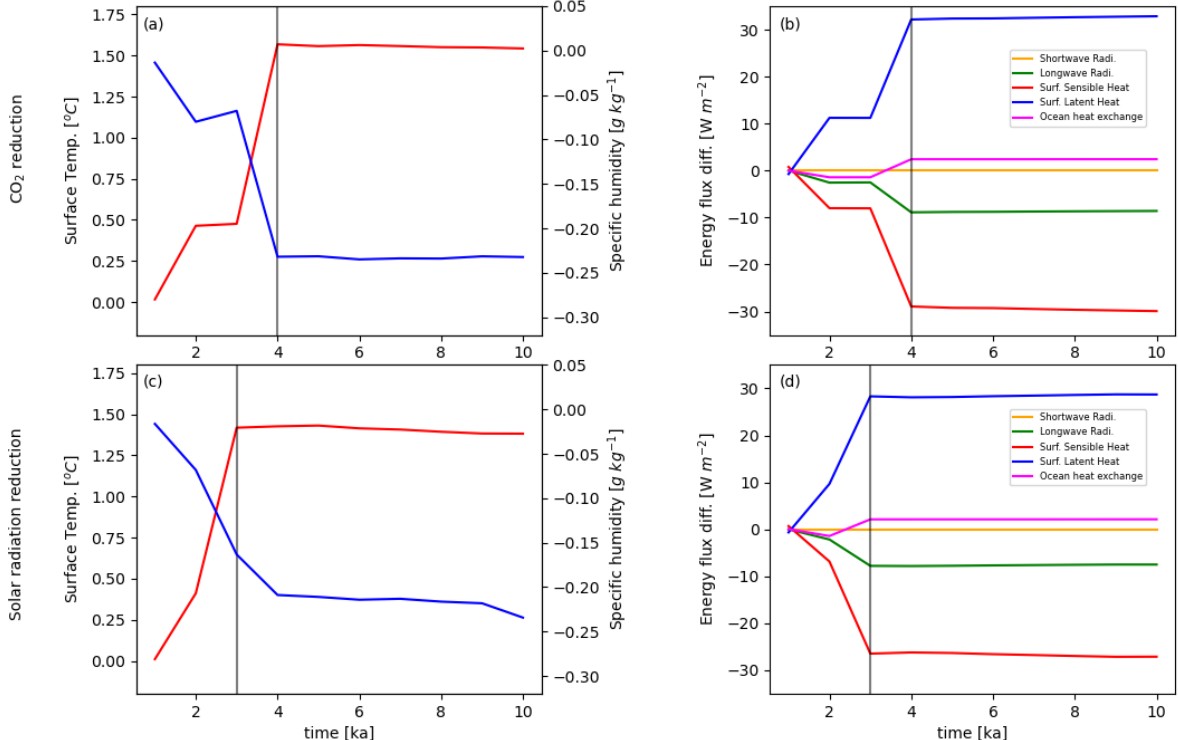

**Fig 19 (a, c) The time evolution of surface temperature response difference (red; unit: ºC) and specific humidity response difference (blue; unit: g kg⁻¹), as well as (b, d) surface energy flux terms response difference (unit: W m⁻²) at a tropical site (140 ºE, 10 ºS). The response difference of variable is defined as using variable response in FULL experiment minus the NSLV experiment. The gray line represents the time point sea-land transition occurs.**

670