# Peer review of "Analysis of the simulated feedbacks on large-scale ice sheets from icesheet climate interactions"

_EGUsphere, 2023_

## Referee Comment (RC1)

**Summary**

The main objective of the paper *"Analysis of the simulated feedbacks on large-scale ice sheets from ice-sheet climate interactions"* by Zhiang Xie and Dietmar Dommenget is to present how different climate/ice sheets feedbacks affect the growth of Northern Hemisphere ice sheets and, how in turn the ice sheets have an impact on the global climate system. To do that the authors use GREB-ISM, a fast coupled climate-ice sheet model that has been presented in a previous study. This paper is therefore a first application of GREB-ISM. The author find that the positive ice-albedo feedback to be the largest among the five that have been assessed. The authors conclude that without this feedback it is impossible to grow such large ice sheets (in the model). An interesting finding.

The scope of this paper is well within the scope of *The Cryosphere*. Climate feedback studies are useful, especially if they tackle such fundamental questions as the build-up of the large Northern Hemisphere ice sheets during the Quaternary.

My main criticism for this paper is the unrealistic nature of the applied forcing (CO2 of 40ppm; solar insolation reduction to 95%), before the authors can even begin to study sensitivities and feedbacks. To me, this suggests that GREB-ISM is just not sensitive enough to (the more realistic) small variations in radiative forcing. Probably a lack of water vapor feedback and lack of realistic atmospheric and oceanic heat transport) to grow large NH ice sheets. I submit that it is difficult to develop a model that captures the important physical processes in a realistic manner, and at the same time computationally fast. The authors should still make sure on which end of the model type spectrum (toy model ↔ fully coupled Earth System model) their model (GREB-ISM) is located. To quote the authors:

> "The simplicity of the model comes with the limitation that the dynamical mean state of the prognostic variables is relatively far away from the observed." (p. 3, L98)

However, I would still recommend this paper for publication, but only after major revisions (see my *General* and *Specific comments* below), because after all the readers and the community shall and will decide about the significance of this study. I know it's a lot of comments, but I hope the authors find value in the suggestions.

Good Luck!
Mario
* * *
**General comments:**

- "Precipitation intensity is often also linked to mountain slopes, as steep topographical changes typically result in heavy precipitation" (p. 2, L40) → It does also dependent on the prevailing wind direction. For example, foehn events lead to drier and warmer conditions on the lee side of a mountain range.
- "ice latent heat" (p. 2, L44) → I would replace the term "ice latent heat" with "latent heat of melting" throughout the text.
- "In addition to the five feedbacks outlined above" (p. 3, L68) → Numbering of the feedbacks would make it clearer for the reader, e.g., as a list.
- "by introducing flux corrections" (p. 3, L99) → Why do you think a flux correction is necessary if you are not running any "realistic" climate simulations anyways?
- "prescribed wind fields" (p. 4, L103) → I can't see how you would be able to study the "ice sheet-topography" feedback in a physically meaningful way.
- "advection and diffusion of heat and moisture is scaled down for increased topography elevation" (p. 5, L143) → Please explain how this is done (equation?); Is there any literature that show how and why this works? E.g., how do you scale down advection?
- "CO2 concentration" (p. 5, L155) → CO2 is an external forcing because you don't account for (bio)geochemistry feedbacks.
- "and solar insolation" (p. 5, L155) → I find this misleading. Quaternary ice age variations are a result of Earth's orbital variations that affect incoming solar radation (and their seasonal distribution). What you are suggesting is to reduce the solar insolation (to 95% of its current value.) This is far from reality, and I can only speculate why you do that: 1) You won't get glacial inception with GREB-ISM. Probably, because it is not sesnitive enough to small variations in insolation. 2) For that reason you also have to reduce CO2 to 40ppm [sic!] (L163), a unrealistic value (for any geological time scale). It's not a typo, is it?
- "A control simulation" (p. 5, L158) → Please include a plot for your control simulation, e.g., for Tsurf.
- "We designed the forcings of the FULL experiments to allow 160 the growth of large-scale Northern Hemispheric continental ice sheets" (p. 5, L160) → What is the equivalent radiative forcing to your CO2/incoming SW reduction? To

me, both experiments are equivalent, if they imply the same radiative forcing. I would therefore suggest to drop one of the two scearios. I would even say that ypour results for both scenarios (and the tested sensitivities) are the same throughout, or, at least I couldn't find any substantial differences in any of the figures and numbers. As a result, you cut your and the readers' time you spent on discussing and contrasting the two scenario in half.

- "five process switches" (p. 6, L168) → It would be useful for the reader to see how the feedbacks enter the model formulation. Please include the relevant equations from the model description paper (e.,g, in an Appendix).
- "a framework to evaluate the feedback strength for ice sheet effect is used in our discussion" (p. 7, L209) → Please, give the reader more background about this feedback framework, as it is the reason for your particular design of your experiments. To quote from said paper (their page 9):

  > "The methodology requires explicitly identifying (1) a perturbation or a class of perturbations, (2) *a response variable* involved in the feedback loop, (3) *the full system* with all processes operating and its response to the perturbation, and (4) the *reference system* with the process of interest *not operating* and the reference system response to the perturbation." [my emphasis]

- "$c$" is the feedback strength" (p. 7, L214) → While you are following the *Goose et al. (2018)* definition of a general feedback I would suggest to use $\gamma$ instead of $c$.
- "global mean of about -7°C in" (p. 8, L236) → Using the temperature response and the applied radiative forcing (reduction), this could be translated into the traditional climate sensitivity. It would be useful to see how your model climate sensitivity compares to other models (and observations).
- "much weaker" (p. 8, L263) → Can you quantify this? You compute dimensionless feedback factors, so I assume there is a way to make them comparable, at least for *ice sheet thickness* as response variable.
- "indicating that this feedback is mostly an amplifying feedback" (p. 8, L265) → What does "mostly" imply here?
- "This suggests that the build-up of the Arctic ice sheets does hinder the formation of a northern central Asian ice sheet." (p. 9) → I find this quite interesting. Is there a way to further investigate the causes of this hinderance?
- "here are coastal points" (p. 9, L284) → I think it would be useful to exclude (or mask) those coastal points from the analysis as they become qualitatively different in their climate response.
- "in more detail in the next section." (p. 10, L302) → I've read this now three or four times. It indicates that something is wrong with the structure of the paper, or of your arguments. Please, help the reader and revise the structure so the readers don't have to jump back and forth.
- "ice transport and ice sheet size feedback" (p. 10)→ If you refer to these terms, please make sure that you introduce them to the reader.
- "Second, the topography feedback in those early studies also included the atmospheric circulation changes, such as stationary wave patterns, which are absent in our study." (p. 10, L326) → I think this is critical and one important reason to **not** include the topography sensitivity in your study.
- "This is an interesting subject that warrants further investigation." (p. 11, L345) → This is an opportunity you should not miss. The albedo representation in your model setup is almost too simple to trust that the feedback has any real meaning. For example, your land albedo is as small as the ocean, but should be in the order of 0.2-0.5 (e.g., bright deserts). Is it worthwile exploring different albedo schemes?
- "longer snowing seasons" (p. 11, L358) → Can you quantify this? E.g., from *X* days to *Y* days.
- "snowfall rates" (p. 11, L359) → Is it larger snowfall rates or accumulated snow throughout the longer winter season?
- "control climate" (p. 12, L379) figure caption says FULL, and I thought control is present-day with no large NH ice sheets.
- "blocking the flow of air across the newly formed mountain ranges" (p. 12, L393) → I would like to see how this blocking looks like in practice. I assume the (u,v) winds have been adjusted (based on something, I can't find in this paper), similar to the "flux corrections", so show the blocking in terms of a vector field. (For example, Fig 5, https://journals.ametsoc.org/view/journals/clim/25/6/jcli-d-11-00218.1.xml)
- "without any other external forcing," (p. 13, L398) → Topography only means that ice sheets are mountains with prescribed land albedo? And only lapse rate (and wind corrections) operating?
- "Further studies with more realistic simulations of changes in the atmospheric and oceanic circulation need to be conducted to better understand the global impact of ice sheets" (p. 14, L444) → This is true in general. But how would you address this problem in your model, specifically?
- In general, this paper could benefit from proof-reading or copy-editing. There is a lot of fluff and unnecessary words (see below for a selection)
* * *
**Specific (or technical) comments:**

- "In study" (p. 1, L8) → "In the study"

- "yrs" (p. 1, L9) → "years"
- "response" (p. 1, L13) response of what? Surface temperature?
- "has" (p. 1, L22) → "have"
- delete "will" (p. 1, L25)
- "model simulations" (p. 1, L26) system using climate model simulations.
- "relation" (p. 1, L31) → "relationship"
- "albedo" (p. 1, L32) → "ice-albedo"
- "snowfall" (p. 2, L35) → just "snow"
- "," (p. 2, L39) → no comma here
- delete "essentially" (p. 2, L50)
- delete "As a result," (p. 3, L82)
- "temperature tendency equation" (p. 6, L189) No such equation is shown.
- "$c_{ALBD} = c_{FULL} - c_{NOALBD}$" (p. 7, L219) → Shouldn't it be: $c_{ALBD} = \frac{c_{FULL} - c_{NALBD}}{c_{FULL}}$, accoring to *Goose et al. (2018)* ?
- "lifting" (p. 8, L242) Please use a different terms, as this could suggest to mean (tectonic up)lifting which it doesn't.
- "gird" (p. 8, L246) → grid
- "," (p. 9, L269) → no comma
- "but" (p. 9, L270) replace with "and"
- "all feedbacks have a direct feedback" (p. 9, L298) → What? Rephrase.
- "all feedbacks have an opposite sign feedback on the surface temperature over remote ice-free regions with varying strength" (p. 9, L299) → This sentence is really confusing and needs reworking. Try to clarify what you want to say here.
- "latter" (p. 10, L314) Not clear if this refers to "weaker for the surface temperature" or "the snowfall feedback" from the previous sentence.
- "as we only consider" (p. 10, L315) replace with "as can be seen in the"
- "significant" (p. 10, L319) What do you mean by "significant"?
- delete "adjacent" (p. 11, L332)
- "abortion" (p. 11, L333) absorption
- "The effect relatively strong in the Arctic" (p. 11, L337) → There is a verb missing: "is"
- delete "conceptually" (p. 11, L339)
- "what has been described" (p. 11, L339) And what is that?
- delet "above physical process of the" (p. 11, L340)
- change "is" to "are" (p. 11, L344)
- change "Snowfall rate" to "Snow" (p. 11, L347)
- delete "Most" (p. 11, L348)
- "local or zonal mean" (p. 11, L350) → Which one is it? Having the equation for precipitation would be useful.
- change "northern hemisphere" to "Northern Hemisphere" (p. 11, L354)
- delete "clear" (p. 11, L355)
- change "decrease" to "decreased" (p. 11, L360)
- delete "The development of the" (p. 11, L361)
- delete "The ice latent heat required to melt ice is substantial." (p. 12, L366)
- delete "substantial" (p. 12, L368)
- "allow the ice sheets to accumulated" (p. 12, L369) → check grammar
- change "sheet" to sheets (p. 12, L382)
- delete "clearly" (p. 12, L388)
- "(Fig. 7i,j)" (p. 12, L393) → I assume you mean Fig. 8.
- "NPREP" (p. 13, L397) → This should be listed in Sect 2.3
- "s" (p. 13, L404) → capital "S".
- change "lowers" to "drops" (p. 13, L406)
- change "bedrock shallower" to "a bathymetry lower" (p. 13, L407)
- "," (p. 13, L414) → no comma
- delete "but" (p. 14, L440)
- change "minor" to "small" (p. 14, L440)
- "This" (p. 14, L442) → What does "this" refer to here? Please, clarify.
- change "does simulate" to "simulates" (p. 14, L442)
- (p. 14, L443) Please add: ", a limitation of the GREB-ISM." to "..., but not in the oceanic heat transport."

- delete "further" (p. 14, L445)
- change "most significant" to "dominant" or "strongest" (p. 14, L445)
- delete "like to be" (p. 14, L447)
- delete "clearly somewhat" (p. 15, L467)
- "does not include all important aspects." (p. 15, L467) → I think you want to say something else, or do you really mean: "The above discussion ... does not include all important aspects."?
- delete "As already mentioned above" (p. 15, L467)
- change "later" to "latter" (p. 15, L472)
- "The GREB-ISM model can address such problems, but may also need further development to address some more complex aspects." (p. 15, L473) This is very vague and unspecific. Delete?
- "mm dy-1." (p. 31, L635) → I don't know what that unit is

---

## Author Comment (AC1)

**Revisions of "Analysis of the simulated feedbacks on large-scale ice sheets from ice-sheet climate interactions"**

Dear Editor and referees,

we like to thank the two referees and editor for the time spend on reviewing this manuscript and for the many very helpful comments they provided. We think the referee comments have helped us to substantially improve the presentation of this work.

With best regards,

Zhiang Xie, Dietmar Dommenget

*Referee #1*

*The main objective of the paper "Analysis of the simulated feedbacks on large-scale ice sheets from ice-sheet climate interactions" by Zhiang Xie and Dietmar Dommenget is to present how different climate/ice sheets feedbacks affect the growth of Northern Hemisphere ice sheets and, how in turn the ice sheets have an impact on the global climate system. To do that the authors use GREB-ISM, a fast coupled climate-ice sheet model that has been presented in a previous study. This paper is therefore a first application of GREB-ISM. The author find that the positive ice-albedo feedback to be the largest among the five that have been assessed. The authors conclude that without this feedback it is impossible to grow such large ice sheets (in the model). An interesting finding.*

*The scope of this paper is well within the scope of The Cryosphere. Climate feedback studies are useful, especially if they tackle such fundamental questions as the build-up of the large Northern Hemisphere ice sheets during the Quaternary.*

*My main criticism for this paper is the unrealistic nature of the applied forcing (CO2 of 40ppm; solar insolation reduction to 95%), before the authors can even begin to study sensitivities and feedbacks. To me, this suggests that GREB-ISM is just not sensitive enough to (the more realistic) small variations in radiative forcing. Probably a lack of water vapor feedback and lack of realistic atmospheric and oceanic heat transport) to grow large NH ice sheets. I submit that it is difficult to develop a model that captures the important physical processes in a realistic manner, and at the same time computationally fast. The authors should still make sure on which end of the model type spectrum (toy model <-> fully coupled Earth System model) their model (GREB-ISM) is located. To quote the authors: "The simplicity of the model comes with the limitation that the dynamical mean state of the prognostic variables is relatively far away from the observed." (p. 3, L98)*

*However, I would still recommend this paper for publication, but only after major revisions (see my General and Specific comments below), because after all the readers and the community shall and will decide about the significance of this study. I know it's a lot of comments, but I hope the authors find value in the suggestions.*

*Good Luck!*

*Mario*

**Response:** Thank you for your thoughtful review and insightful comments. We respond to

each of the comments below. We also clarify in our responses to the other comments that the model has a realistic mean state, due to the flux correction terms and a realistic climate sensitivity, including a realistic water vapor feedback. We also better explain the motivation for the solar radiation forcing, arguing that, although idealized, it is somewhat realistic.
* * *
**General comments:**

*"Precipitation intensity is often also linked to mountain slopes, as steep topographical changes typically result in heavy precipitation" (p. 2, L40) -> It does also dependent on the prevailing wind direction. For example, foehn events lead to drier and warmer conditions on the lee side of a mountain range.*

**Response:** That is correct. Now the sentence has been modified as "Precipitation intensity is often also linked to mountain slopes, as steep topographical changes typically result in heavy precipitation over upwind slopes" in main text.
* * *
*"ice latent heat" (p. 2, L44) -> I would replace the term "ice latent heat" with "latent heat of melting" throughout the text.*

**Response:** Thanks for your suggestion. "ice latent heat" has been replaced with "latent heat of melting" while "ice latent heat feedback" has been replaced with "ice melting latent heat feedback".
* * *
*"In addition to the five feedbacks outlined above" (p. 3, L68) -> Numbering of the feedbacks would make it clearer for the reader, e.g., as a list.*

**Response:** In response, the sentence now reads as follows:
"In addition to the five feedbacks outlined above, which we will mainly discuss in this paper and listed in Table 1, there are several other feedbacks associated with the climate-ice sheet interaction."

*"by introducing flux corrections" (p. 3, L99) -> Why do you think a flux correction is necessary if you are not running any "realistic" climate simulations anyways?*

**Response:** We are running "*realistic*" simulations, as each simulation is related to a control simulation that is closed to today's mean climate. Without flux corrections this would not be the case and would strongly alter the outcomes not only on a regional scale. We are also arguing that our response experiments, although idealized, are not entirely unrealistic. See also response to the other points. We have added a sentence in the model description to better highlight that the flux correction will ensure a realistic mean state in the control simulations.

*"prescribed wind fields" (p. 4, L103) -> I can't see how you would be able to study the "ice sheet-topography" feedback in a physically meaningful way.*
*"advection and diffusion of heat and moisture is scaled down for increased topography elevation" (p. 5, L143) ->Please explain how this is done (equation?); Is there any literature that show how and why this works? E.g., how do you scale down advection?*

**Response:** We have now included an appendix explaining all important equations of the GREB-ISM. We hope this now clarifies how the topography affects different processes, including the transport of heat and moisture.
While the GREB-ISM does not have complex atmospheric circulation changes as they would be simulated in GCMs, it does have some simulation of changes in atmospheric transports. Given that the literature on AGCM simulations on these time scales are rare or non-existing, we think that the discussion of the GREB-ISM results is relevant and can give a first order approximation against which future more complex simulations with AGCMs can be compared.

*"CO2 concentration" (p. 5, L155) -> CO2 is an external forcing because you don't account for (bio)geochemistry feedbacks.*

*"and solar insolation" (p. 5, L155) -> I find this misleading. Quaternary ice age variations are a result of Earth's orbital variations that affect incoming solar radation (and their seasonal distribution). What you are suggesting is to reduce the solar insolation (to 95% of its current value.) This is far from reality, and I can only speculate why you do that: 1) You won't get glacial inception with GREB-ISM. Probably, because it is not sesnitive enough to small variations in insolation. 2) For that reason you also have to reduce CO2 to 40ppm [sic!] (L163), a unrealistic value (for any geological time scale). It's not a typo, is it?*

**Response:** The reviewer comments make it clear that we have not well motivated our solar and $CO_2$ forcing experiments. We therefore revised section 2.3 (Design of sensitivity experiments) to better argue for the forcings and explain the motivations.

In short summary, solar insolation variations over the past million years are in the order of 20 W m$^{-2}$ for a 24 hrs mean in summer for the higher latitudes, which corresponds roughly to about 5% of the solar constant. This is based on analysis of data from Huybers and Eisenman (2006). See also Abe-Ouchi et al. (2013). However, these solar radiation variations are not globally uniform, but have complex meridional and seasonal patterns that are different at different time scales. To simplify the experiments we conduct a -5% solar radiation reduction scenario.

Since the GREB model does not consider the carbon cycle, we can for the purpose of these experiments consider both, the solar and atmospheric $CO_2$ variations, as external forcings, and focus on understanding the climate-ice sheet feedback in the presence of solar insolation and $CO_2$ forcings.

In the $CO_2$ reduction scenario, the $CO_2$ concentration drops from 340 ppm in the control to 40 ppm in the scenario, which is not something that has been observed in the past million years, but was chosen to mimic a global mean response similar to the solar radiation reduction scenario. The separation of the solar and atmospheric $CO_2$ in these two scenarios allow us to analysis potential differences in the climate response to the difference forcing agents. Both scenarios present a relatively strong forcing, but are not entirely unrealistic in amplitude of response. Both scenarios allow the growth of large-scale Northern Hemispheric continental ice sheets, which is important for the analysis of ice-sheet feedbacks.

Reference:

Abe-Ouchi, A., Saito, F., Kawamura, K., Raymo, M. E., Okuno, J., Takahashi, K. and Blatter, H.: Insolation-driven 100,000-year glacial cycles and hysteresis of ice-sheet volume, Nature, 500(7461), 190–193, doi:10.1038/nature12374, 2013.

Huybers, P., & Eisenman, I. (2006). Integrated summer insolation calculations. IGBP PAGES/World Data Center for Paleoclimatology Data Contribution Series, 79.
* * *
*"A control simulation" (p. 5, L158) -> Please include a plot for your control simulation, e.g., for Tsurf.*

**Response:** We have now included a figure in the appendix to show the annual mean Tsurf and ice distribution (Fig. A1).
* * *
*"We designed the forcings of the FULL experiments to allow the growth of large-scale Northern Hemispheric continental ice sheets" (p. 5, L160) -> What is the equivalent radiative forcing to your CO2/incoming SW reduction? To me, both experiments are equivalent, if they imply the same radiative forcing. I would therefore suggest to drop one of the two scearios. I would even say that your results for both scenarios (and the tested sensitivities) are the same throughout, or, at least I couldn't find any substantial differences in any of the figures and numbers. As a result, you cut your and the readers' time you spent on discussing and contrasting the two scenario in half.*

**Response:** We like to keep both scenarios, as it is not immediately clear that they would result into the same response. Solar radiation forcing and $CO_2$ forcing do have regionally different forcing strength (e.g. solar is stronger in the tropics), and the feedbacks discussed in this study are also potentially different for different forcing types (e.g. ice-albedo feedback). Since, both forcings are active during ice-age cycles, we think the discussion is relevant, even though the response differences are small. This as such is an important result.

*"five process switches" (p. 6, L168) -> It would be useful for the reader to see how the feedbacks enter the model formulation. Please include the relevant equations from the model description paper (e.,g, in an Appendix).*

**Response:** We have included additional explanations and relevant equations related to the physical processes and switches in the appendix.
* * *
*"a framework to evaluate the feedback strength for ice sheet effect is used in our discussion" (p. 7, L209) -> Please, give the reader more background about this feedback framework, as it is the reason for your particular design of your experiments. To quote from said paper (their page 9): "The methodology requires explicitly identifying (1) a perturbation or a class of perturbations, (2) a response variable involved in the feedback loop, (3) the full system with all processes operating and its response to the perturbation, and (4) the reference system with the process of interest not operating and the reference system*
*response to the perturbation." [my emphasis]*
*" c" is the feedback strength" (p. 7, L214) -> While you are following the Goose et al. (2018) definition of a general feedback I would suggest to use $\gamma$ instead of c.*

**Response:** Our reference to Goose et al. was misleading. We are not following Goose et al. in detail. We follow a simple linear energy balance equation, as also discussed in many other studies. We now cite Forster and Gregory (2006) as an example.
* * *
*"global mean of about -7°C in" (p. 8, L236) -> Using the temperature response and the applied radiative forcing (reduction), this could be translated into the traditional climate sensitivity. It*

*would be useful to see how your model climate sensitivity compares to other models (and observations).*

**Response:** We have now added a sentence for the climate sensitivity of the GREB model. The global surface temperature change at equilibrium for a doubling of $CO_2$, is found to be 2.85°C, which is similar to most CMIP6 models.

Given that the forcings applied in our experiments are very different from a doubling of $CO_2$, we think it is not helpful to state an abstract climate sensitivity value, but rather state the global mean temperature change for the given forcing, as we do in the manuscript.
* * *
*"much weaker" (p. 8, L263) -> Can you quantify this? You compute dimensionless feedback factors, so I assume there is a way to make them comparable, at least for ice sheet thickness as response variable.*

**Response:** This part mainly focused on the zonal mean ice thickness change. The "much weaker" refers to less reduction from FULL experiment of zonal mean ice thickness in HEAT experiment compared with ALBD experiment (Figure 7b, d). The text in manuscript has been updated:

"While the ice melting latent heat feedback is also a positive effect, its impact is considerably weaker compared to the ice-albedo feedback. This is evident in the simulation results, where the ice melting latent heat feedback leads to ice sheet response of several hundred meters near 70°N (Fig 7b, d), whereas the ice-albedo feedback results in virtually no ice sheets."
* * *
*"indicating that this feedback is mostly an amplifying feedback" (p. 8, L265) -> What does "mostly" imply here?*

**Response:** We revised the text to better explain this. It was related to most regions.
* * *
*"This suggests that the build-up of the Arctic ice sheets does hinder the formation of a northern central Asian ice sheet." (p. 9) -> I find this quite interesting. Is there a way to further investigate the causes of this hinderance?*

**Response:** In the manuscript, we have conducted an extra experiment to explore this process in section 4.4. Given the length of the manuscript, we have to leave further analysis to future work, which we suggested also in the final discussion section.
* * *
*"there are coastal points" (p. 9, L284) -> I think it would be useful to exclude (or mask) those coastal points from the analysis as they become qualitatively different in their climate response.*

**Response:** We think that it would be better to include them to give a full global climate discussion. The "qualitatively different" is relevant for these locations.
* * *
*"in more detail in the next section." (p. 10, L302) -> I've read this now three or four times. It indicates that something is wrong with the structure of the paper, or of your arguments. Please, help the reader and revise the structure so the readers don't have to jump back and forth.*

**Response:** Apologies for the confusion caused by the repeated reference to "detail in the next section." Upon revisiting the structure of the paper, we have made adjustments to provide a more coherent flow of information for the readers. We have addressed the ice sheet blocking effect in section 2.2 and connected it to the detailed experiment conducted in section 4.4:
"The ice sheet blocking effect induces opposite anomalies in the regions north and south of the ice sheets. A comprehensive analysis of this feedback will be presented in section 4.4."
The relevant repeated section jumps are deleted.

Additionally, the reference to "detail in the next section" related to global sea level change has been removed as it is redundant.
* * *
*"ice transport and ice sheet size feedback" (p. 10)-> If you refer to these terms, please make sure that you introduce them to the reader.*

**Response:** We have simplified the statement to avoid confusion. We now only mention ice transport, which is also introduced in the GREB model description.
* * *
*"Second, the topography feedback in those early studies also included the atmospheric circulation changes, such as stationary wave patterns, which are absent in our study." (p. 10, L326) -> I think this is critical and one important reason to **not** include the topography sensitivity in your study.*

**Response:** While it is true that our study does not incorporate atmospheric circulation changes and stationary wave patterns associated with topography feedback, it is important to note that these are not the sole factors contributing to the overall impact. This includes changes in the surface temperature by increased elevation, reduced humidity and related precipitation, and reduced atmospheric transports of heat and moisture. Given that the literature on AGCM simulations on these time scales are rare or non-existing, we think that the discussion of the GREB-ISM results is relevant and can give a first order approximation against which future, more complex simulations with AGCMs can be compared.

We have now included an appendix explaining all important equations of the GREB-ISM. We hope this now clarifies how the topography affects different processes, including the transport of heat and moisture. This illustrates that the GREB-ISM is indeed sensitive to topographic changes.
* * *
*"This is an interesting subject that warrants further investigation." (p. 11, L345) -> This is an opportunity you should not miss. The albedo representation in your model setup is almost too simple to trust that the feedback has any real meaning. For example, your land albedo is as small as the ocean, but should be in the order of 0.2-0.5 (e.g., bright deserts). Is it worthwile exploring different albedo schemes?*

**Response:** We indeed agree that this is an important aspect that needs further investigation. We do highlight this in the final summary section and suggest further studies.
* * *
*"longer snowing seasons" (p. 11, L358) -> Can you quantify this? E.g., from X days to Y days.*
**Response:** We have now quantified this. The increase are up to 80 dyas in the first 20yrs of the scenario simulation.
* * *
*"snowfall rates" (p. 11, L359) -> Is it larger snowfall rates or accumulated snow throughout the longer winter season?*

**Response:** It is an increase in accumulated snowfall, which we now stated in the manuscript.
* * *
*"control climate" (p. 12, L379) figure caption says FULL, and I thought control is present-day with no large NH ice sheets.*

**Response:** We apologize for the confusion caused by the mistake in the figure caption. You are correct that the term "control climate" refers to the present-day climate with no large

Northern Hemisphere ice sheets. The figure caption has been updated to accurately reflect the experimental setup.
* * *
*"blocking the flow of air across the newly formed mountain ranges" (p. 12, L393) -> I would like to see how this blocking looks like in practice. I assume the (u,v) winds have been adjusted (based on something, I can't find in this paper), similar to the "flux corrections", so show the blocking in terms of a vector field. (For example, Fig 5,*

*[https://journals.ametsoc.org/view/journals/clim/25/6/jcli-d-11-00218.1.xml](https://journals.ametsoc.org/view/journals/clim/25/6/jcli-d-11-00218.1.xml))*

**Response:** We have now included an appendix explaining all important equations of the GREB-ISM. We hope this now clarifies how the topography affects different processes, including the transport of heat and moisture.

There are a number of processes that affect the atmospheric transport. They are related to equations A2, A4, A9 and A10.
* * *
*"without any other external forcing," (p. 13, L398) -> Topography only means that ice sheets are mountains with prescribed land albedo? And only lapse rate (and wind corrections) operating?*

**Response:** That is correct. In the topography-only experiment described in the paper (Figure 17a, b), we manipulate the surface elevation ($z_{topo}$; see model equations in the appendix) of the grids to represent the presence of ice sheets as mountains. The modifications in surface elevation lead to alterations in various factors, including surface temperature, diffusion rate, and precipitation, which are influenced by the new mountainous topography. This experiment allows us to isolate the effect of topography changes on the climate system.

We have now included the appendix with the model equations to better illustrate what the model is simulating.

*"Further studies with more realistic simulations of changes in the atmospheric and oceanic circulation need to be conducted to better understand the global impact of ice sheets" (p. 14, L444) -> This is true in general. But how would you address this problem in your model, specifically?*

**Response:** In our current version of GREB-ISM v1.0, fully dynamic-coupled atmospheric and oceanic circulation is not included. However, there are alternative approaches that can be employed to address this limitation. One possible method in the GREB-ISM is to incorporate another prescribed wind field derived from the Last Glacial Maximum (LGM) and assume linear changes in the meridional and zonal wind fields from the present-day conditions to the LGM conditions based on global sea level changes. This approach would require additional adjustments to the flux correction scheme and the implementation of benchmark experiments. While these developments are beyond the scope of our current study, we acknowledge the importance of considering more realistic simulations of circulation changes and will explore these possibilities in future. However, we do not want to add more discussions in the manuscript, to keep the discussion short.

*In general, this paper could benefit from proof-reading or copy-editing. There is a lot of fluff and unnecessary words*
*(see below for a selection)*

**Response:** We have carefully worked through the manuscript again to improve the presentation.

Specific (or technical) comments:

*"In study" (p. 1, L8) -> "In the study"*

**Response:** Done.
* * *
*"yrs" (p. 1, L9) -> "years"*

**Response:** Done.
* * *
*"response" (p. 1, L13) response of what? Surface temperature?*

**Response:** We revised it to "response of the climate system ".
* * *
*"has" (p. 1, L22) -> "have"*

**Response:** Done.
* * *
*delete "will" (p. 1, L25)*

**Response:** Done.
* * *
*"model simulations" (p. 1, L26) system using climate model simulations.*

**Response:** We revised the sentence.
* * *
*"relation" (p. 1, L31) -> "relationship"*

**Response:** Done.
* * *
*"albedo" (p. 1, L32) -> "ice-albedo"*

**Response:** Done.
* * *
*"snowfall" (p. 2, L35) -> just "snow"*

**Response:** Done.
* * *
*"," (p. 2, L39) -> no comma here*

**Response:** To better express the idea, the sentences have been changed as: "The snowfall feedback is closely linked to the topography feedback, as the decrease in precipitation due to surface temperature drop is also influenced by the elevated surface height of ice sheets."
* * *
*delete "essentially" (p. 2, L50)*

**Response:** Done.
* * *
*delete "As a result," (p. 3, L82)*

**Response:** Done.
* * *
*"temperature tendency equation" (p. 6, L189) No such equation is shown.*

**Response:** We have now included an appendix in which the equation is show and this is now referenced in this section.
* * *
*"$c_{ALBD} = c_{FULL} - c_{NOALBD}$" (p. 7, L219) → Shouldn't it be:*

*$c_{ALBD} = \frac{c_{FULL} - c_{NALBD}}{c_{FULL}}$, accoring to Goose et al. (2018) ?*

**Response:** The reference to Goose et al. has been misleading. We follow a simple linear energy balance equation, as also discussed in many other studies. We now cite Forster and Gregory (2006) as an example.
* * *
*"lifting" (p. 8, L242) Please use a different terms, as this could suggest to mean (tectonic up)lifting which it doesn't.*

**Response:** "topography lifting" is replaced with "surface elevation rise".
* * *
*"gird" (p. 8, L246) -> grid*

**Response:** Done.
* * *
*"," (p. 9, L269) -> no comma*

**Response:** Done.
* * *
*"but" (p. 9, L270) replace with "and"*

**Response:** Done.
* * *
*"all feedbacks have a direct feedback" (p. 9, L298) -> What? Rephrase.*

**Response:** We revised this section.
* * *
*"all feedbacks have an opposite sign feedback on the surface temperature over remote ice-free regions with varying strength" (p. 9, L299) -> This sentence is really confusing and needs reworking. Try to clarify what you want to say here.*

**Response:** We revised this section.
* * *
*"latter" (p. 10, L314) Not clear if this refers to "weaker for the surface temperature" or "the snowfall feedback" from the previous sentence.*

**Response:** To avoid misunderstanding, it has been changed to: "The comparable strength of ice melting latent heat feedback and snowfall feedback is an interesting finding."

*"as we only consider" (p. 10, L315) replace with "as can be seen in the"*

**Response:** Done.
* * *
*"significant" (p. 10, L319) What do you mean by "significant"?*

**Response:** We revised the sentence and now state "There is a clear topography feedback for the ice volume …".
* * *
*delete "adjacent" (p. 11, L332)*

**Response:** We revised the sentence to better highlight the local and global effects.
* * *
*"abortion" (p. 11, L333) absorption*

**Response:** Done.
* * *
*"The effect relatively strong in the Arctic" (p. 11, L337) -> There is a verb missing: "is"*

**Response:** Added now.
* * *
*delete "conceptually" (p. 11, L339)*

**Response:** Done.
* * *
*"what has been described" (p. 11, L339) And what is that?*

**Response:** More detail has been included now: "In general, the ice-albedo feedback in our simulations is conceptually similar to what has been described in previous studies, where the increase in ice cover leads to an increase in surface albedo, resulting in decreased absorption of solar radiation and subsequent cooling (Fyke et al., 2018; Willeit and Ganopolski, 2018)."

*delet "above physical process of the" (p. 11, L340)*

**Response:** Done.
* * *
*change "is" to "are" (p. 11, L344)*

**Response:** Done.
* * *
*change "Snowfall rate" to "Snow" (p. 11, L347)*

**Response:** Done.
* * *
*delete "Most" (p. 11, L348)*

**Response:** Done.
* * *
*"local or zonal mean" (p. 11, L350) -> Which one is it? Having the equation for precipitation would be useful.*

**Response:** It is both and we revised the text. We have now included the model equations in the appendix, which better illustrates how precipitation is calculated.
* * *
*change "northern hemisphere" to "Northern Hemisphere" (p. 11, L354)*

**Response:** Done.
* * *
*delete "clear" (p. 11, L355)*

**Response:** Done.
* * *
*change "decrease" to "decreased" (p. 11, L360)*

**Response:** Done.

*delete "The development of the" (p. 11, L361)*

**Response:** Done.
* * *
*delete "The ice latent heat required to melt ice is substantial." (p. 12, L366)*

**Response:** Done.
* * *
*delete "substantial" (p. 12, L368)*

**Response:** Done.
* * *
*"allow the ice sheets to accumulated" (p. 12, L369) -> check grammar*

**Response:** Now it has been changed as: "In the context of the seasonal cycle, it plays a crucial role in overcoming the warm summer season and facilitating the accumulation of ice sheets from one winter to the next."
* * *
*change "sheet" to sheets (p. 12, L382)*

**Response:** Done.
* * *
*delete "clearly" (p. 12, L388)*

**Response:** Done.
* * *
*"(Fig. 7i,j)" (p. 12, L393) -> I assume you mean Fig. 8.*

**Response:** Yes, you are right. Sorry for the confusion and it has been corrected.
* * *
*"NPREP" (p. 13, L397) -> This should be listed in Sect 2.3*

**Response:** This was a typo. And it should be "NSNOW".
* * *
*"s" (p. 13, L404) -> capital "S".*

**Response:** Done.
* * *
*change "lowers" to "drops" (p. 13, L406)*

**Response:** Done.
* * *
*change "bedrock shallower" to "a bathymetry lower" (p. 13, L407)*

**Response:** Done.
* * *
*"," (p. 13, L414) -> no comma*

**Response:** Done.
* * *
*delete "but" (p. 14, L440)*

**Response:** Done.
* * *
*change "minor" to "small" (p. 14, L440)*

**Response:** Done.
* * *
*"This" (p. 14, L442) -> What does "this" refer to here? Please, clarify.*

**Response:** The sentence has been changed to "However, the ocean circulation can also modify the remote influence."
* * *
*change "does simulate" to "simulates" (p. 14, L442)*

**Response:** Done.
* * *
*(p. 14, L443) Please add: ", a limitation of the GREB-ISM." to "..., but not in the oceanic heat transport."*

**Response:** Done.
* * *
*delete "further" (p. 14, L445)*

**Response:** Done.
* * *
*change "most significant" to "dominant" or "strongest" (p. 14, L445)*

**Response:** Done.
* * *
*delete "like to be" (p. 14, L447)*

**Response:** Done.
* * *
*delete "clearly somewhat" (p. 15, L467)*

**Response:** Done.
* * *
*"does not include all important aspects." (p. 15, L467) -> I think you want to say something else, or do you really mean: "The above discussion ... does not include all important aspects."?*

**Response:** It has been changed: "The above discussion of feedbacks, while providing valuable insights, is idealized and does not encompass all important aspects."
* * *
*delete "As already mentioned above" (p. 15, L467)*

**Response:** Done.
* * *
*change "later" to "latter" (p. 15, L472)*

**Response:** Done.
* * *
*"The GREB-ISM model can address such problems, but may also need further development to address some more complex aspects." (p. 15, L473) This is very vague and unspecific. Delete?*

**Response:** It has been deleted.
* * *
*"mm dy-1." (p. 31, L635) -> I don't know what that unit is*

**Response:** It is millimeter per day. They are all changed to "mm d$^{-1}$" in both manuscript and figures.
* * *
*This study uses a simplified coupled ice sheet-climate model to analyze feedback between large-scale ice sheets and the climate system. The climate part is a global energy balance model with an invariant wind field. The ice sheet model has four vertical layers and a positive degree day scheme to calculate the surface mass balance. One of the main findings is that the albedo feedback dominates ice growth. This is not groundbreaking in itself but I welcome the author's approach to take advantage of a computationally tool to systematically test the sensitivity of the ice-climate system.*

*The scope of the manuscript is interesting and well-suited for The Cryosphere (although to me Climate of the Past is an even better fit). My criticism focusses on the limitations of the GREB model and the extent to which they are tested and discussed in the manuscript. I think these aspects must be addressed before publication.*

**Response:** Thank you for your valuable comments and suggestions. We revised our manuscript to address the reviewers point in respect to the limitations of the GREB model and the aims of this study. Please, see our response to the specific comments below.
* * *
*1) It is difficult to understand to what degree the results depend on model limitations. If I understood correctly, GREB uses a prescribed and time-invariant wind field. This average field and (invariant) statistics about its variance are then used to prescribe moisture transport. This is a strong limitation as previous studies have found the dynamic response of, e.g., the stationary wave pattern (Löfverström and Liakka, 2016) or local circulation changes around ice sheets (Merz et al., 2014a,b) to be very important. Also, the ocean circulation in GREB cannot change, which is another strong limitation. I understand that testing the assumptions that went into making the efficient model can only be tested fully in a more complex model, but it should be possible to estimate some simplifications by adjusting model parameters within GREB. As an example, there must be a parameterization for meridional heat transport by the ocean that could be changed to approximate changes in the circulation. Such changes are believed to be essential for climate-ice sheet interactions on longer time scales. Similarly, the lack of a dynamic response of the atmosphere to the growth of the Laurentide ice sheet*

*should be tested. How important are the feedbacks in the presence of this additional effect? This needs to be quantified.*

**Response:** The reviewer is correct in pointing out the limitations of the GREB-ISM model regarding the prescribed and time-invariant wind field and the absence of dynamic changes in ocean circulation. It is indeed difficult to evaluate the limitations of the GREB model, given the lag of observations or more complex model simulations.

In this study, our aim was to present a series of sensitivity experiments focusing on different feedbacks. It is indeed valid to look at model parameter variations and how they affect the results, but given the length of the current manuscript, we think this is beyond what can be done within this one study.

We tried to improve the introduction of the GREB model and its limitations. We further discussed the results carefully and pointed out potential limitations. It is the nature of such simplified model simulations that they can only give a first guess. In the final section we discuss a number of avenues on how this study should be continued to address the model limitations and potential model parameter uncertainties (e.g. ice albedo, ocean/atmosphere circulation).
* * *
*2) Related to this first point, I would like to see a more detailed discussion of the merits that the author's approach holds. Why should simulations with a strongly simplified model be considered by journals and their readers? How can these simple models answer questions that more sophisticated models cannot? Does the manuscript in its present form really take advantage of the low computational cost of GREB-ISM? I do not think so as it appears to only present a handfull of simulations, each only representing a few hours of time on a regular single CPU. Why were not more changes in GHGs tested? Why not different changes in solar irradiation? Also, what does a 5% reduction in solar output mean for simulations that run over 100,000 years? Is this a constant offset? Does it have seasonality? GREB-ISM can, and maybe should, be used to do more comprehensive tests of Milankovitch forcing, including the importance of time scales (obliquity, precession, etc.).*

**Response:** We have tried to better introduce the GREB model and highlight its usefulness for this study. In our manuscript, we have conducted over 14 pairs of fully transient sensitivity

experiments using GREB-ISM, with each about 100,000 yrs long. In total these are more than 1.5 mill. yrs. of simulations, which would be challenging to accomplish with more computationally expensive models. The total time cost for these experiments exceeds more than 10 days, even in our simplified model setting. Therefore, while the computational cost is low compared to more complex models, we have made efficient use of GREB-ISM to isolate and analyze specific feedback processes, offering valuable insights into the ice sheet-climate interactions.

Regarding the suggestion of exploring more comprehensive tests of Milankovitch forcing, we agree that it is an interesting topic for future investigation. However, our current study aims to provide a basic concept of how the Earth system responds to solar and greenhouse gas forcing in a global uniform way. The consideration of Milankovitch forcing introduces additional complexities and non-uniformities to the system, which deserves a dedicated analysis in a separate study. We appreciate the suggestion and will keep it in mind for future research.
* * *
*Minor comments:*

*- The manuscript requires language editing. I think that running individual paragraphs through GPT or similar would probably solve 95% of the issues.*

**Response:** A proofreading has been done and we corrected some parts with help of GPT.
* * *
*line 138: "Atmospheric blocking" usually refers to a specific type of circulation anomaly, different from what is meant here.*

**Response:** It has been changed to "ice sheet blocking effect".
* * *
*line 179: Most (all?) of the processes described here cannot be addressed with GREB-ISM*

**Response:** The GREB-ISM is able to simulate these topography and snowfall feedback processes. The detail of physical process and relevant equations for each switch in the GREB-ISM is now included in the appendix.
* * *
*figure 6: This only presents anomalies. How does the ice topography of FULL without perturbations look?*

**Response:** The results of the control run have been included in the appendix (Figure A1).
* * *
*figure 10: The division by 10 for one of the columns must be immediately visible in the figure without the need to read the caption.*

**Response:** Thanks for your suggestion. We now use "$\frac{1}{10}$ALBD" in the xlabel to indicate the downscaling manufacturing process.
* * *
*figure 12: What does "mm/dy" mean? Per day (d) or per year (yr)?*

**Response:** It is millimeter per day. They are all changed to "mm d$^{-1}$" in both manuscript and figures.
* * *
*figure 14: I found this figure and the corresponding text difficult to follow and cannot say I am convinced.*

**Response:** We revised the discussion of the text, the organization of the figure and the figure caption to improve the presentation.
* * *
*figures in general: I think there is too many figures for the relatively straightforward point that the manuscript is trying to make.*

**Response:** We have included multiple figures to provide a comprehensive and robust presentation of our findings. Each figure serves a specific purpose in illustrating different aspects of our analysis and helps support the main points of the manuscript. We believe that the inclusion of these figures enhances the clarity and rigor of our study.
* * *
*Literature (if not yet included in the manuscript):*

*Merz et al. 2014b: https://doi.org/10.1002/2014JD021940*

*Merz et al. 2014a: https://doi.org/10.5194/cp-10-1221-2014*

**Response:** Done.

---

## Author Comment (AC2)

[revised manuscript text omitted]

In addition to the five feedbacks outlined above, which we will mainly discuss in this paper and listed in Table 1, there are several other feedbacks associated with the climate-ice sheet interaction. Pollard et al. (2015) pointed out that Antarctic Ice Sheet potentially retreated because of hydrofracturing and ice cliff failure, caused by the ice shelf basal melting. And both Abe-Ouchi et al. (2013) and Han et al. (2021) suggested the feedbacks associated with the solid Earth deformation takes effect during the ice sheet evolution. Moreover, Willeit and Ganopolski (2018) discussed the strong influence from surface albedo, and concluded that the existence of dust on surface snow potentially causes a large uncertainty in model simulation.

Due to high computational expense of long-term simulation, the relevant ice sheet-climate interaction studies commonly use simple Earth system models dynamically coupled to an ice sheet model with low resolution and strong parameterizations (Abe-Ouchi et al., 2013; Fyke et al., 2011; Ganopolski et al., 2010; Tigchelaar et al., 2019). This method is very close to the fully coupled simulation and thus shows much more details on the climate-ice sheet interaction. Most of those studies focused on the simulation of real ice age cycles driven by combination of greenhouse gas and orbital forcing. However, we also need idealised sensitivity experiments to help us better understand the role of ice sheet-climate interaction.

In this study, we will conduct a series of idealised sensitivity experiments with GREB-ISM climate model (Xie et al., 2022), to explore ice sheet-climate interaction and associated physical processes. The GREB-ISM model is a fast and simple fully coupled climate model that resolves the global climate system on a coarse resolution grid. The model allows a simple deconstruction of the different elements in the climate system, while at the same time has realistic representations of important elements of the climate system. While the model does not include complex atmospheric or ocean circulation dynamics or the carbon cycle, it does simulate the five main feedback processes discussed above.

The study is organised as follows: Our experiment design and related procedure will be explained in section 2. The findings of our sensitivity experiment will be summarised in section 3, along with a comparison of different feedbacks. The section 4 will go through each of the five feedback processes and discuss the mechanisms controlling each feedback. The final section, will summarize and discuss the study.

**2 Model and method**

**2.1 Model**

In this study, we use the Globally Resolved Energy Balance - Ice Sheet Model v1.0 ( GREB-ISM v1.0 , Xie et al., 2022), in which ice sheets and ice shelves are simulated on a global grid, fully interacting with the climate simulation of surface temperature, snowfall, albedo, land-sea mask, topography, and sea level. Thus, it is a fully coupled atmosphere, ocean, land and ice sheet model. The GREB-ISM climate components consist of three physical layers, including atmosphere, surface and sub-surface ocean with horizontal resolution of $3.75^o$ x $3.75^o$ (96 x 48 points). The model equations relevant for this study are presented in the appendix.

In addition to the prognostic equations of the temperature of the three layers the GREB-ISM model is also simulating the atmospheric water vapor as a prognostic equation based on evaporation, precipitation and atmospheric transport. Thus, precipitation is simulated in the GREB-ISM model as a result of the atmospheric state, following a simple diagnostic model by Stassen et al., (2019). The equilibrium climate sensitivity (ECS), which measures the global warming at equilibrium for a doubling of $CO_2$, for GREB v1.0, the climate component of GREB-ISM, is found to be 2.85°C (Nicholls et al., 2020). This value falls within the range of ECS values reported by CMIP6 model members, which range from 1.8°C to 5.6°C (Meehl et al., 2020).

The simplicity of the model comes with the limitation that the dynamical mean state of the prognostic variables is relatively far away from the observed. This is addressed in the GREB-ISM by introducing flux corrections to the tendency equations of the surface temperature and the atmospheric humidity to enforced a monthly climatology as observed for presented day external forcings (e.g. $CO_2$-concentraions and solar radiation). As a result, the control simulations in the GREB-ISM model experiments discussed in this study closely resemble the mean climate state of the present day (see Fig. A1).

Unlike atmospheric general circulation models, the GREB model does not simulate variations in the atmospheric circulation (weather), but has prescribed wind fields. Subsequently, the model has no variability other than the response to external forcings. As a consequence, the model state of a single year can be used as an estimate of the climate state for a given forcing at the given time. No long-term time averaging is needed.

The ice sheet model part, simulates the prognostic equation for the ice thickness on the same horizontal grid as the GREB model, including energy-based surface mass balance, ice temperature on 4 vertical layers, ice sheet dynamics (transport) and floating ice shelf processes (Xie et al., 2022). The ice sheet mass balance accumulates the simulated precipitation as snowfall rates, when both the surface and air temperatures are below or near the freezing point. The GREB-ISM v1.0 is able to simulate the 100,000 model years within 24 hrs wall time on a standard personal computer, making long term simulations and a large set of sensitivity experiments feasible.

**2.2 Simulated processes related to the ice sheet-climate interactions**

The five feedbacks in our discussion interact strongly with one another through a number of physical mechanisms. For the following discussions in the results section, it is important to explain some physical effects simulated in the GREB-ISM regarding the formation of an ice sheet and the interaction with the climate state. We have outlined several key mechanisms:

*Ice-albedo effect*

The GREB-ISM albedo scheme is different from the original GREB. In the original GREB model (without ice sheet simulation) surface albedo is diagnosed as function of surface temperature (Fig 1a). In the GREB-ISM the albedo is a function of ice thickness (Fig 1b) following Eq. A15.

*Snowfall effect*

In the GREB-ISM model precipitation is diagnosed on the basis of atmospheric, local and zonal mean humidity, relative humidity, mean and standard deviation of vertical atmospheric velocities (Eq. A11). The latter two are external boundary conditions and the first two are simulated based on the state of the climate system. Thus, precipitation rates adjust to the climate state. Over land the precipitation rate are further flux corrected to match the observed precipitation rate to ensure realistic mass balances for continental ice sheets (Xie et al., 2022). This flux correction is designed in a way that the precipitation rates over land will still be able to change proportional to the zonal mean atmospheric humidity.

*Elevation-surface temperature effect*
Higher elevated surfaces have colder temperatures, as they are in sensible heat exchange with an atmosphere that is colder than lower elevated surfaces, due to the adiabatic lapse rate in the troposphere. This effect is commonly included in ice sheet modelling and plays critical role for ice sheet development (Albrecht et al., 2020; Fyke et al., 2011). Although the GREB-ISM has only a single layer atmosphere, the surface temperature decreases with elevation due to sensible heat exchange with the atmosphere, following a fixed dry adiabatic lapse rate (0.6K per 100m; Eq. A8).

*Ice sheet blocking effect*
Large continental ice sheets affect the atmospheric circulation, blocking air flow across the ice sheets. In early studies, air transport was frequently associated with ice sheet growth via circulation changes caused by the topography change (Felzer et al., 1996; Herrington and Poulsen, 2012). The GREB-ISM model simulates heat and moisture transport in the atmosphere but assumes prescribed atmospheric mean wind conditions that do not respond to changes in the topography (Eqs. A2 and A4). However, the GREB-ISM simulation of advection and diffusion of heat and moisture is scaled down for increased topography elevation, effectively reducing atmospheric transport across high topography, thus responding to the formation of large ice sheets (Eqs. A6 and A10). The ice sheet blocking effect induces opposite anomalies in the regions north and south of the ice sheets. A comprehensive analysis of this feedback will be presented in section 4.4.

*Global sea level change*
The global volume of grounded ice sheets affects the global sea level in the GREB-ISM. This change in sea level affects the elevation of land topography, the grounding lines of ice shelves and the land-sea mask for the climate (Eq. A6). Figure 2 shows the control bedrock, illustrating regions with shallow ocean points that will be most affected by sea level changes. An ocean point that turns into a land point, is assumed to have a reduced soil moisture, and therefore has strongly reduced surface evaporation. It also has no more interactions with the subsurface ocean layer. Both effects will affect the surface temperature heat balance by reducing latent heat flux and subsurface ocean cooling (Eq. A1).

**2.3 Design of sensitivity experiments**

The most prevalent external forcings in the last million years are variations in the solar insolation due to changs in the orbital parameters (Tabor et al., 2015). This external forcing is accompanied by an internal positive climate feedback from the carbon cycle, which amplifies the climate response to the external forcing by elevating atmospheric $CO_2$ concentrations during warmer periods (Ganopolski and Brovkin, 2017). Since the GREB model does not consider the carbon cycle, for the purpose of these experiments, we can treat both solar variations and atmospheric $CO_2$ variations as external forcings. This allows us to concentrate on examining the climate-ice sheet feedback in the context of solar insolation and $CO_2$ forcings.

The solar insolation variations over the past million years are in the order of 20 $W m^{-2}$ for a 24 hrs mean in summer for the higher latitudes, which corresponds roughly to about 5% of the solar constant. However, these solar radiation variations are not globally uniform, but have complex meridional and seasonal patterns that are different at different time scales. To simplify the experiments, we conduct two idealised scenarios: a 5% solar radiation reduction scenario with unchanged atmospheric $CO_2$ concentration (340ppm) and a $CO_2$ reduction scenario, in which the solar radiation is unchanged. In the $CO_2$ reduction scenario, the $CO_2$ concentration decreases from 340 ppm in the control to 40 ppm in the scenario. Although this specific $CO_2$ concentration drop has not been observed in the past million years, it was chosen in order to simulate a global mean response similar to the solar radiation reduction scenario. The separation of the solar and atmospheric $CO_2$ in these two scenarios allow us to analyse potential differences in the climate response to the different forcing agents. Both scenarios present a relatively strong forcing, but are not entirely unrealistic in amplitude of response. Both scenarios allow the growth of large-scale Northern Hemispheric continental ice sheets, which is important for the analysis of ice-sheet feedbacks.

A control simulation and a scenario simulation are included in each of our experiments. We first perform a 30 kyr control simulation with current $CO_2$ concentrations and solar radiation (control simulation results are provided in Fig. A1), then a 100 kyr scenario simulation. In total we analysis more than 1.5 million years of simulations.

To evaluate the sensitivity of the climate system to different processes, we conducted additional sensitivity experiments, in which elements of the climate system are turned "off", following the approach in Dommenget and Floeter (2011). In the GREB-ISM, we created five process switches (albedo, latent heat of melting, topography, snowfall, and sea level feedback) to examine the effect of each climate-ice sheet feedback (Table 1). The relevant equations are detailed in Appendix. Here is a brief introduction:

ALBD (albedo feedback) switch: The surface albedo of a snow/ice-covered points is higher than those of ice-free points (Eq. A15). As a result, the surface albedo is a function of ice thickness in the GREB-ISM. If we turn off the ALBD switch, the surface albedo will be unaffected by changes in the ice thickness and will remain the same as in the control run (including its seasonal variations).

TOPO switch (topography feedback): In the GREB-ISM, surface elevation depends on a constant bedrock elevation and the simulated ice thickness (Eq. A6). Surface elevation changes affect the atmospheric humidity, precipitation, sensible heat and

Deleted: This external forcing is combined with an internal positive climate feedback from the carbon cycle that substantially enhances the climate response to the external forcing by increasing the atmospheric $CO_2$ concentration, when the climate is warmer
Deleted: we can for the purpose of these experiments consider bBoth, the solar and atmospheric $CO_2$ variations, as external forcings, and focus have the ability to alter the Earth's net radiation budget and thus induce global warming or cooling. As a result, the focus of experiments in this study will be on understanding the climate-ice sheet feedback in the presence of $CO_2$ and solar insolation and $CO_2$ forcings
Moved (insertion) [1]
We have
Moved up [1]: We have two scenarios: a $CO_2$ reduction

[revised manuscript text omitted]
. While the ice melting latent heat feedback is also a positive effect, its impact is considerably weaker compared to the ice-albedo feedback. This is evident in the simulation results, where the ice melting latent heat feedback leads to ice sheet response of several hundred meters near 70°N (Fig 7b, d), whereas the ice-albedo feedback results in virtually no ice sheets. The regional pattern of the response difference is similar to the overall response in the FULL experiment, indicating that the amplifying feedback at different locations is of similar strength, not altering the regional characteristics.

The snowfall feedback is the only significant negative feedback, which is of similar strength to the ice melting latent heat feedback, but with opposite sign. The snowfall feedback has a tendency to shift the ice sheet further north, which may partially be related to the larger land masses to the south of the ice sheets that form in the NSNOW experiment. The snowfall feedback is also the only feedback which has a significant influence on the Antarctic ice sheets.

The topography feedback is weaker than the above-mentioned feedbacks, and it is mostly a positive feedback. The sea level feedback in turn is more complex, as it tends to shift ice sheets in location rather than just amplifying or damping ice sheets growth. The sea level reduction due to the global growth of grounded ice sheets turns some of the shallow ice shelves in the

Arctic Ocean to land points, which allows the growth of large ice sheets. This shifts the ice sheets from Asia towards the Arctic Ocean. Here it is interesting to note that the ice sheet growth is not only amplified over the Arctic Ocean, but also damped over northern central Asia. This suggests that the build-up of the Arctic ice sheets does hinder the formation of a northern central Asian ice sheet.

445

Figures 8 and 9 show the response of the surface temperature in the FULL experiment and the response difference of the sensitivity experiments in reference to the FULL experiment. Starting with the albedo feedback, we can first of all note that the surface temperature response difference (Fig. 8c,d) is very similar to the FULL experiment response (Fig. 8a,b), indicating that most of the regional differences in the surface temperature response results from the ice-albedo feedback. This is also

450 illustrated in Fig. 9a,c where we can see that the NALBD experiment response is fairly similar on all latitudes, in contrast to the strong variations seen in the FULL experiments. The ice-albedo feedback is not only affecting the ice-covered regions, but does affect all latitudes (Fig. 9b), due to the atmospheric interactions (e.g., transport of heat and moisture). In contrast to this global positive feedback, there are coastal points in which the ice-albedo feedback is negative (including the ice-albedo feedback leads to less cooling in response to the forcings; Fig. 8c,d). This results from the global sea level change.

455 The snowfall feedback is a strong negative feedback over the regions with ice sheet growth, which directly results from the cooling effect of the surface elevation lifting by the ice sheets. Interestingly, lower latitudes see a positive feedback from the precipitation, similar in strength to the ice-albedo feedback (Fig.9 b,d), which is related to the ice sheet blocking effect. This aspect is similar for the topography feedback, but in general with opposite sign and somewhat weaker. The switch in sign of the feedback is also following closely the regions where the ice sheets grow (compare Figs. 6 and 8).

460 The ice melting latent heat feedback for the surface temperature is also following a similar pattern as the precipitation and topography feedbacks. While the direct response over the regions with ice sheet growth is similar in strength, it is weaker in the regions far away from ice covered zones. The same holds for the sea level feedback. Here we see positive feedbacks over ice covered regions, following the ice sheet changes. In addition, we see negative feedbacks over ice-free regions similar to those in the NHEAT experiment.

465

In general, we find that all feedbacks directly influence the surface temperature over regions experiencing ice sheet growth. This is attributed to the cooling effect caused by the lifting of surface elevation by the ice sheets. While some feedbacks are positive and others are negative over regions experiencing ice sheet growth, they all exhibit an reversed sign feedback on the surface temperature over remote ice-free regions, with varying strength attributed to the ice sheet blocking effect (e.g., positive

470 feedbacks turn into negative feedbacks, and negative feedbacks turn into positive feedbacks). The exception is the ice-albedo feedback, which is positive globally.

We can quantify the strength for each feedback based on Eq. (3) with $\Delta\Phi$ as the global mean ice sheet volume or surface temperature response. The results for each experiment and for both scenarios are shown in Figure 10. The overall feedback strength the for FULL experiments is negative in both scenarios and both variables, as expected for stable global climate

475 system. We can further note that the sum of all the one-switch-off experiments does not add up to the feedback strength of the FULL experiment. This is expected, since not all feedback processes have been considered in these five experiments. For

instance, the Plank radiative feedback and ice transport feedback (Fyke et al., 2018) are essential negative feedbacks that have not been evaluated in our experiments.

The ice-albedo feedback is by far the most important feedback from the five feedbacks discussed for both variables. It is, however, more dominant for the ice sheet volume than for the surface temperatures. The ice melting latent heat feedback is the second strongest feedback for the ice volume, but is somewhat weaker for the surface temperature, where the snowfall feedback is about as strong.

The comparable strength of ice melting latent heat and snowfall feedback is an interesting finding, since the snowfall feedback has significant opposite sign feedbacks over ice-free regions, which the ice melting latent heat feedback does not have as strongly. This reduces the overall strength of the snowfall feedback as can be seen in the global mean values in Figure 10. Since the snowfall feedback is the only significant negative feedback over ice sheet regions, the NSNOW experiment has by far the largest extent in continental ice sheets. This affects the linear approximation of Eq. (3).

There is a clear topography feedback for the ice volume, but near zero for the surface temperature, as the feedback has opposite signs for different regions. Similarly, the sea level feedback is overall fairly weak, which is also due to its counteracting effect by shifting the location of ice sheets.

Our results here agree with numerous earlier investigations, which established that albedo feedback has a dominating role in Northern Hemisphere climate change during paleoclimate (Abe-Ouchi et al., 2007; Budyko, 1969; Felzer et al., 1996). However, there are some differences between our study and the previous works. First, those early studies only evaluate the effect from the ice-albedo and topography feedback and few of them discussed the ice latent heat, snowfall and sea level feedbacks. Second, the topography feedback in those early studies also included the atmospheric circulation changes, such as stationary wave patterns, which are absent in our study.

**4 Analysis of the feedback processes**

We now focus on the individual feedback processes in more detail, aiming at getting a more in depth understanding of how theses feedbacks work and what is causing some of the unexpected characteristics.

**4.1 Albedo feedback**

High surface albedo is an essential feature of snow-covered area. As a result, local surface temperature drops due to less shortwave radiation absorption, and remote regions cool due to reduced atmospheric heat transport. The process of the albedo feedback is quite clear. The initial surface cooling caused by solar radiation or $CO_2$ reduction favors ice sheet formation, which in turn further increases the surface albedo in most of Northern Hemisphere land by more than 5% (Fig 11a, c). The brighter ice surface leads to more than 10 W m$^{-2}$ annual surface solar radiation reduction in the whole Northern Hemisphere land, facilitating the initial surface cooling from the external forcing (Fig 10b, d). The effect is relatively strong in the Arctic, as the albedo changes are in summer, while it is not as strong in lower latitudes, as here the albedo changes are mostly in winter (not shown). In general, the ice-albedo feedback in our simulations is similar to what has been described in previous studies, where

the increase in ice cover leads to an increase in surface albedo, resulting in decreased absorption of solar radiation and subsequent cooling (Fyke et al., 2018; Willeit and Ganopolski, 2018).

The ice-albedo feedback does not quite explain why the FULL experiment has such a strongly amplified surface temperature response for large ice sheets (compare Figs. 7 and 9). This is related to the surface elevation lifting or the topography feedback, which will be discussed further below. It is worth noting that we did not systematically evaluate albedo scheme changes (e.g., changes in the albedo of ice). Willeit and Ganopolski (2018) pointed out that the Northern Hemisphere ice sheets over the last glacial cycle are very sensitive to the representation of snow albedo in the modelling. This is an interesting subject that warrants further investigation.

**4.2 Snowfall feedback**

Snowfall is the primary source of ice mass in GREB-ISM. When the surface and air temperatures drop near or below the freezing point, precipitation turns to snow in the GREB-ISM. Previous studies linked the snowfall feedback directly to topography changes (Fyke et al., 2018; Hakuba et al., 2012; Löfverström and Liakka, 2016; Medley and Thomas, 2019). In the GREB-ISM, the precipitation is simulated based on the state of the atmosphere and is related to local and zonal mean atmospheric water vapor capacity (see section 2.2), so it is indirectly connected with surface temperature and also the topography.

The cooler climate in the scenario runs lead to reductions in the precipitation rates that affect the snowfall rates in different ways. Figure 12 shows the changes in the snowfall rates in the Northern Hemisphere for the winter and summer seasons in the FULL experiment. We can see a reduction in snowfall rates in higher latitudes in winter and increases in snowfall rates in lower latitudes (Fig. 12a). The reduction at higher latitudes results from the cooler atmospheric temperatures that reduce the atmospheric water vapor concentrations, and therefore also reduce the precipitation and snowfall rates. At lower latitudes this is also the case, but at the same time the cooler temperatures in lower latitudes allow longer snowing seasons, with increases of up to 80 days in the first 20 yrs of the scenario run. This leads to increase in accumulated snowfall over winter (Fig. 12a). This effect dominates in summer also at higher latitudes (Fig. 12b).

The combined effect of decreased snowfall in winter and increased snowfall in summer at higher latitudes leads to an overall decrease in the annual mean snowfall rates (Fig. 13) reducing the ice sheets growth. The large continental ice sheets further reduces the snowfall rates, due to the lifting of the surface elevation, which reduces the precipitation (Fig. 12c,d). In summary, the snowfall rates are decreasing during the ice sheet formations, due to the colder climate and the surface elevation lifting. This is a negative feedback for the ice sheet growth.

**4.3 Ice melting latent heat feedback**

To melt a 1 m ice column requires $3.03 \times 10^8$ J m$^{-2}$ energy, which is roughly equivalent to warm the land surface by 60$^{\circ}$C. This additional heat required to change the surface temperature of ice covered regions does provide an additional inertia to the climate system. It will in general delay the climate response to heating. In the context of the seasonal cycle, it plays a crucial role in overcoming the warm summer season and facilitating the accumulation of ice sheets from one winter to the next.

We can illustrate this effect by considering a location in northern central Asia (90 °E, 70 °N) with seasonally varying ice cover in the control simulation, see Figure 14. At the start of the FULL experiment (first decade), the melting of ice in summer has a cooling effect (black lines in Fig. 14a,d) allowing ice sheet to stay longer during spring, reducing the positive degree days (PDD; Fig. 14b,e). This effect allows the ice sheet over the next decades to grow fast than those in the NHEAT experiment, which do not have this seasonal effect (Fig.14c,f). As a result, more surface shortwave radiation is reflected during the warm

605 season (Fig 14a, d), further reducing the temperature (Fig 14c, f). After multiple repetitions, it ends up with a permanent ice cover zone.

This seasonal effect of ice melting latent heat feedback works in regions with the short melting seasons, where the decreasing melting season length eventually causes irreversible ice accumulation (Fig. 15). First, we can note that the regions around the

610 arctic have a short season of positive degree days (PDD) in the control climate (Fig. 15a). These are also the regions where the differences in the warm season temperatures and build up of ice sheets in the first decades are largest between the FULL and the NHEAT simulations (Fig. 15b-e). In the FULL simulation, including the ice melting latent heat feedback, the summer temperatures decrease faster and the ice sheets build up more strongly.

**4.4 Topography feedback**

615 In our sensitivity experiments, following a significant cooling caused by $CO_2$ and solar radiation decrease, large ice sheets start growing, eventually raising the surface elevation to thousands of metres (Fig 6). This causes the surface temperature to cool down as it is now exchanging a sensible heat flux with an ambient atmosphere that is much cooler due to the lapse rate in the troposphere (see section 2.2). Figure 16 shows how the surface temperature and the surface elevation change relate to each other. It follows the fix dry adiabatic lapse rate used in the GREB model. Thus, most of the strong cooling over the large ice

620 sheets is a result of the tropospheric lapse rate cooling. This feedback is widely discussed and also referred to as "elevation-surface mass balance feedback" (Abe-Ouchi et al., 2007; Edwards et al., 2014; Fyke et al., 2018), which is a positive feedback during deglaciation period (Abe-Ouchi et al., 2013; Kapsch et al., 2021).

The development of large continental ice sheets also affects the atmospheric circulation, blocking the flow of air across the newly formed mountain ranges. This blocking effect can be noted in the surface temperature response changes (Fig. 8i,j),

625 where surface temperature poleward of the ice sheets is cooler in the FULL experiment than in the NTOPO experiment and warmer equatorward. It can also be seen in the other sensitivity experiments (Fig. 8), but with different strength or sign, depending on the nature of the feedback under consideration.

We conducted an additional sensitivity experiment in which we prescribed only the change in topography of [NSNOW] – [FULL] without any other external forcing (changes in $z_{topo}$ only; see model equations in the appendix), 
[revised manuscript text omitted]

Our study aims to assess the strength of various ice sheet-climate feedbacks within the context of a simplified model and under a basic external forcing. By utilizing this simplified model, we are able to conduct a series of sensitivity experiments, allowing

720    for the isolation and analysis of specific physical processes. The above discussion of feedbacks, while providing valuable insights, is idealized and does not encompass all relevant aspects. A more detailed studied of how atmospheric and ocean circulation changes due to the build-up of ice sheets is important for a better understanding of the global impact of ice sheets.

This was also recognized as important factors in previous studies (Felzer et al., 1996; Larour et al., 2012). We further also neglected additional feedbacks such as ice sheets transport or interaction with the carbon cycle that can have impacts on the global climate system. Finally, the simulation conducted here only considered the build-up of ice sheets, but did not consider a melting phase. Studying the latter will also be important to fully understand the ice sheets feedbacks.

**6 Appendix**

The GREB-ISM model equations relevant to this study are shortly presented in this section.

All symbols and parameters are listed in Table A1. There are five equations for prognostic variables in the GREB-ISM: surface temperature ($T_{surf}$), atmospheric temperature ($T_{atmos}$), subsurface ocean temperature ($T_{ocean}$), surface humidity ($q_{air}$), and ice thickness ($H$):

$$\gamma_{surf} \frac{dT_{surf}}{dt} = F_{solar} + F_{thermal} + F_{latent} + F_{sense} + F_{ocean} + F_{ice} + F_{correct} \qquad (A1)$$

$$\gamma_{atmos} \frac{dT_{atmos}}{dt} = -F_{sense} + Fa_{thermal} + Q_{latent} + \rho_a \cdot \gamma_{atmos} \left( \kappa_a \nabla^2 T_{atmos} - \vec{V} \cdot \nabla T_{atmos} \right) \qquad (A2)$$

$$\frac{dT_{ocean}}{dt} = \frac{1}{\Delta t} \Delta T o_{entrain} - \frac{1}{\gamma_{ocean} - \gamma_{surf}} F o_{sense} + F o_{correct} \qquad (A3)$$

$$\frac{dq_{air}}{dt} = \Delta q_{eva} + \Delta q_{precip} + \rho_v \cdot \left( \kappa_a \nabla^2 q_{air} - \vec{V} \cdot \nabla q_{air} \right) + \Delta q_{correct} \qquad (A4)$$

$$\frac{\partial H}{\partial t} = s - a - \nabla \cdot \left( \vec{V}_i H \right) \qquad (A5)$$

A few of the climate process relevant for the experiments are detailed as follows:

**Flux corrections**

The flux correction terms $F_{correct}$, $Fo_{correct}$ and $\Delta q_{correct}$ in Eqs. A1, A3 and A4 are allowing the GREB-ISM simulations to have control climate simulations close to todays climate, see Fig. A1.

**Topographic effects**

The topography is diagnosed as:

$$z_{topo} = b + H \qquad \qquad \text{if } b + r_\rho H > 0 \text{ (grounded grids)}$$

$$z_{topo} = (1 - r_\rho)H \qquad \text{if } b + r_\rho H < 0 \text{ and } H > 10\ m \text{ (ice shelf)} \tag{A6}$$

$$z_{topo} = -0.1\ m \qquad \text{otherwise (ocean grids)}$$

where bedrock $b$ is modified based on the sea level change. $z_{topo}$ influences a number of processes in the GREB-ISM model. The sensible heat ($F_{sense}$) exchange between the atmospheric layer and the surface is:

$$F_{sense} = ct_{sense}(T_{asurf} - T_{surf}) \tag{A7}$$

where the surface air temperature $T_{asurf}$ is related to the air temperature ($T_{atmos}$) scaled by the topography with a lapse rate ($\Gamma$):

$$T_{asurf} = T_{atmos} + \Gamma \cdot z_{topo} \tag{A8}$$

The saturated humidity ($q_{sat}$) for estimating $\Delta q_{eva}$ and $F_{latent}$ is:

$$q_{sat} = e^{\frac{-z_{topo}}{z_{vapor}}} \cdot 3.75 \cdot 10^{-3} \cdot e^{\frac{17.08085 \cdot (T_{surf} - 273.15)}{T_{surf} - 273.15 + 234.175}} \tag{A9}$$

The transport by advection and diffusion in Eqs. A2 and A4 are scaled by the topography:

$$\rho_a = e^{\frac{-z_{topo}}{z_{air}}} \qquad \text{and} \qquad \rho_v = e^{\frac{-z_{topo}}{z_{vapor}}} \tag{A10}$$

**Precipitation**

Precipitation ($p$) is diagnosed as:

$$p = r_{precip} \cdot \left( q_{air} \cdot \left( c_{rq} \cdot rq + c_\omega \cdot \omega_{mean} + c_{\omega SD} \cdot \omega_{SD} \right) + q_{zonal} \cdot p_{correct} \right) \tag{A11}$$

Here the correction term $p_{correct}$ is only applied over land points.

**Snowfall**

The snowfall ($s$) is directly linked with precipitation in the GREB-ISM:

$$s = \frac{r}{r_\rho} p \tag{A12}$$

where $r$ is a snowfall temperature scaling rate:

$$r = \begin{cases} 1, & T_{asurf} < T_m \text{ and } T_{surf} < T_m - 2^oC \\ \frac{1}{2}\left(1 - \frac{T_{surf} - T_m}{2^oC}\right), & T_{asurf} < T_m \text{ and } T_m - 2^oC < T_{surf} < T_m + 2^oC \\ 0, & otherwise \end{cases} \tag{A13}$$

**Surface albedo**

The absorbed solar radiation ($F_{solar}$) is directly related with surface albedo ($\alpha_{surf}$) by equation:

$$F_{solar} = (1 - \alpha_{clouds}) \cdot (1 - \alpha_{surf}) \cdot S_0 \cdot r_{solar} \tag{A14}$$

where surface albedo ($\alpha_{surf}$) is a function of ice thickness ($H$):

$$\begin{aligned} \alpha_{surf} &= 0.1 & H &= 0.0 \\ \alpha_{surf} &= 0.1 + 17.5 \text{ m}^{-1} \cdot H & H &\in [0.0, 0.02 \text{ m}] \\ \alpha_{surf} &= 0.45 & H &> 0.02 \text{ m} \end{aligned} \tag{A15}$$

**7 Code availability**

[revised manuscript text omitted]

Morlighem, M., Williams, C. N., Rignot, E., An, L., Arndt, J. E., Bamber, J. L., Catania, G., Chauché, N., Dowdeswell, J. A.,

925 Dorschel, B., Fenty, I., Hogan, K., Howat, I., Hubbard, A., Jakobsson, M., Jordan, T. M., Kjeldsen, K. K., Millan, R., Mayer, L., Mouginot, J., Noël, B. P. Y., O'Cofaigh, C., Palmer, S., Rysgaard, S., Seroussi, H., Siegert, M. J., Slabon, P., Straneo, F., van den Broeke, M. R., Weinrebe, W., Wood, M. and Zinglersen, K. B.: BedMachine v3: Complete Bed Topography and Ocean Bathymetry Mapping of Greenland From Multibeam Echo Sounding Combined With Mass Conservation, Geophys. Res. Lett., 44(21), 11,051-11,061, doi:10.1002/2017GL074954, 2017.

930 Morlighem, M., Rignot, E., Binder, T., Blankenship, D., Drews, R., Eagles, G., Eisen, O., Ferraccioli, F., Forsberg, R., Fretwell, P., Goel, V., Greenbaum, J. S., Gudmundsson, H., Guo, J., Helm, V., Hofstede, C., Howat, I., Humbert, A., Jokat, W., Karlsson, N. B., Lee, W. S., Matsuoka, K., Millan, R., Mouginot, J., Paden, J., Pattyn, F., Roberts, J., Rosier, S., Ruppel, A., Seroussi, H., Smith, E. C., Steinhage, D., Sun, B., Broeke, M. R. va. den, Ommen, T. D. va., Wessem, M. van and Young, D. A.: Deep glacial troughs and stabilizing ridges unveiled beneath the margins of the Antarctic ice sheet, Nat. Geosci., 13(2),

935 132–137, doi:10.1038/s41561-019-0510-8, 2020.

Nicholls, R. J. Z., Meinshausen, M., Lewis, J., Gieseke, R., Dommenget, D., Dorheim, K., Fan5, C. S., Fuglestvedt, J. S., Gasser, T., Goluke, U., Goodwin, P., Hartin, C., P. Hope, A., Kriegler, E., J. Leach, N., Marchegiani, D., A. McBride, L., Quilcaille, Y., Rogelj, J., J. Salawitch, R., Samset, B. H., Sandstad, M., N. Shiklomanov, A., B. Skeie, R., J. Smith, C., Smith, S., Tanaka, K., Tsutsui, J. and Xie, Z.: Reduced Complexity Model Intercomparison Project Phase 1: Introduction and

940 evaluation of global-mean temperature response, Geosci. Model Dev., 13(11), 5175–5190, doi:10.5194/gmd-13-5175-2020, 2020.

Patel, A., Goswami, A., Dharpure, J. K., Thamban, M., Sharma, P., Kulkarni, A. V. and Oulkar, S.: Estimation of mass and energy balance of glaciers using a distributed energy balance model over the Chandra river basin (Western Himalaya), Hydrol. Process., 35(2), 1–22, doi:10.1002/hyp.14058, 2021.

945 Petit, J. R., Jouzel, J., Raynaud, D., Barkov, N. I., Barnola, J.-M., Basile, I., Bender, M., Chappellaz, J., Davis, M., Delaygue, G., Delmotte, M., Kotlyakov, V. M., Legrand, M., Lipenkov, V. Y., Lorius, C., PÉpin, L., Ritz, C., Saltzman, E. and Stievenard, M.: Climate and atmospheric history of the past 420,000 years from the Vostok ice core, Antarctica, Nature, 399(6735), 429–

436, doi:10.1038/20859, 1999.

Pollard, D. and DeConto, R. M.: Modelling West Antarctic ice sheet growth and collapse through the past five million years, Nature, 458(7236), 329–332, doi:10.1038/nature07809, 2009.

Pollard, D., DeConto, R. M. and Alley, R. B.: Potential Antarctic Ice Sheet retreat driven by hydrofracturing and ice cliff failure, Earth Planet. Sci. Lett., 412, 112–121, doi:10.1016/j.epsl.2014.12.035, 2015.

Ryan, J. C., Smith, L. C., Van As, D., Cooley, S. W., Cooper, M. G., Pitcher, L. H. and Hubbard, A.: Greenland Ice Sheet surface melt amplified by snowline migration and bare ice exposure, Sci. Adv., 5(3), 1–11, doi:10.1126/sciadv.aav3738, 2019.

Schoof, C.: Ice sheet grounding line dynamics: Steady states, stability, and hysteresis, J. Geophys. Res. Earth Surf., 112(3), 1–19, doi:10.1029/2006JF000664, 2007.

Stassen, C., Dommenget, D. and Loveday, N.: A hydrological cycle model for the Globally Resolved Energy Balance (GREB) model v1.0, Geosci. Model Dev., 12(1), 425–440, doi:10.5194/gmd-12-425-2019, 2019.

Tabor, C. R., Poulsen, C. J. and Pollard, D.: How obliquity cycles powered early Pleistocene global ice-volume variability, Geophys. Res. Lett., 42(6), 1871–1879, doi:10.1002/2015GL063322, 2015.

Tigchelaar, M., Timmermann, A., Friedrich, T., Heinemann, M. and Pollard, D.: Nonlinear response of the Antarctic Ice Sheet to late Quaternary sea level and climate forcing, Cryosphere, 13(10), 2615–2631, doi:10.5194/tc-13-2615-2019, 2019.

Willeit, M. and Ganopolski, A.: The importance of snow albedo for ice sheet evolution over the last glacial cycle, Clim. Past, 14(5), 697–707, doi:10.5194/cp-14-697-2018, 2018.

Xie, Z., Dommenget, D., McCormack, F. S. and Mackintosh, A. N.: GREB-ISM v1.0: A coupled ice sheet model for the Globally Resolved Energy Balance model for global simulations on timescales of 100\,kyr, Geosci. Model Dev., 15(9), 3691–3719, doi:10.5194/gmd-15-3691-2022, 2022.

Zeitz, M., Reese, R., Beckmann, J., Krebs-Kanzow, U. and Winkelmann, R.: Impact of the melt-albedo feedback on the future evolution of the Greenland Ice Sheet with PISM-dEBM-simple, Cryosphere, 15(12), 5739–5764, doi:10.5194/tc-15-5739-2021, 2021.

975 **Table 1 Climate – ice sheet feedback and process switches.**

| Feedback | ON | OFF |
| --- | --- | --- |
| Albedo (ALBD) | Albedo change due to ice thickness | Albedo fixed as initial condition |
| Latent heat (HEAT) | Latent heat of melting is negative when melting | Latent heat of melting set as zero |
| Topography (TOPO) | Topography change due to ice thickness, which is related to surface sensible heat, precipitation and heat transportation | Topography fixed as initial condition |
| Snowfall (SNOW) | Snowfall based on local precipitation | Snowfall fixed as initial condition |
| Sea level (SLV) | Sea level change due to global ice volume change | Sea level set as zero |

**Table A1 Symbols and Parameters List**

| variable name | symbol | dimensions | value/unit |
|---|---|---|---|
| ablation rate | $a$ | x, y, t | $m\ s^{-1}$ |
| bed rock elevation | $b$ | x, y, t | $m$ |
| precipitation parameter for relative humidity | $c_{rq}$ | constant | $Pa^{-1}s$ |
| sensible heat bulk coefficient | $ct_{sense}$ | constant | $22.5\ W\ m^{-2}\ K^{-1}$ |
| precipitation parameter for vertical velocity | $c_\omega$ | constant | $Pa^{-1}s$ |
| precipitation parameter for standard deviation of vertical velocity | $c_{\omega SD}$ | constant | $Pa^{-2}s^2$ |
| net longwave radiation for $T_{atmos}$ | $Fa_{thermal}$ | x, y, t | $W\ m^{-2}$ |
| surface flux correction | $F_{correct}$ | x, y, t | $W\ m^{-2}$ |
| ice latent heat flux | $F_{ice}$ | x, y, t | $W\ m^{-2}$ |
| latent heat flux | $F_{latent}$ | x, y, t | $W\ m^{-2}$ |
| land-sea heat difference | $F_{ocean}$ | x, y, t | $W\ m^{-2}$ |
| ocean heat flux correction | $Fo_{correct}$ | x, y, t | $W\ m^{-2}$ |
| sensible heat flux between ocean and surface | $Fo_{sense}$ | x, y, t | $W\ m^{-2}$ |
| sensible heat flux between air and surface | $F_{sense}$ | x, y, t | $W\ m^{-2}$ |
| solar radiation | $F_{solar}$ | x, y, t | $W\ m^{-2}$ |
| net longwave radiation for $T_{surf}$ | $F_{thermal}$ | x, y, t | $W\ m^{-2}$ |
| ice thickness | $H$ | x, y, t | $m$ |
| precipitation | $p$ | x, y, t | $m\ s^{-1}$ |
| precipitation correction | $p_{correct}$ | x, y, t | $kg\ kg^{-1}\ s^{-1}$ |
| latent heat flux in air | $Q_{latent}$ | x, y, t | $W\ m^{-2}$ |
| air specific humidity | $q_{air}$ | x, y, t | $kg\ kg^{-1}$ |
| saturated humidity | $q_{sat}$ | x, y, t | $kg\ kg^{-1}$ |

Formatted Table

| | | | |
|---|---|---|---|
| zonal specific humidity mean | $q_{zonal}$ | x, y, t | $kg\ kg^{-1}$ |
| snowfall temperature scaling rate | $r$ | x, y, t | unitless |
| ratio of ice density to water density | $r_\rho$ | constant | unitless |
| relative humidity | $rq$ | x, y, t | unitless |
| Mean lifetime of water vapour | $r_{precip}$ | constant | $kg\ kg^{-1}\ s^{-1}$ |
| ratio of absorbed solar insolation at atmosphere top | $r_{solar}$ | y, t | unitless |
| solar radiation from space | $S_0$ | y, t | $W\ m^{-2}$ |
| ice accumulation rate (snowfall) | $s$ | x, y, t | $m\ s^{-1}$ |
| air temperature | $T_{atmos}$ | x, y, t | K |
| surface air temperature | $T_{asurf}$ | x, y, t | K |
| ice melting temperature | $T_m$ | x, y, t | K |
| ocean temperature | $T_{ocean}$ | x, y, t | K |
| surface temperature | $T_{surf}$ | x, y, t | K |
| horizontal wind speed | $\vec{V}$ | x, y | $m\ s^{-1}$ |
| ice flow horizontal velocity (vertical mean) | $\vec{V}_i$ | x, y, t | $m\ s^{-1}$ |
| the reference air mass height | $z_{air}$ | constant | 8000 m |
| surface topography | $z_{topo}$ | x, y, t | m |
| the reference maximum height that water vapor can reach | $z_{vapor}$ | constant | 5000 m |
| cloud albedo | $\alpha_{clouds}$ | x, y | unitless |
| surface albedo | $\alpha_{surf}$ | x, y, t | unitless |
| lapse rate | $\Gamma$ | constant | $-0.006\ K\ m^{-1}$ |
| heat capacity of atmosphere layer | $\gamma_{atmos}$ | x, y, t | $J\ K^{-1}\ m^{-2}$ |
| heat capacity of ocean layer | $\gamma_{ocean}$ | x, y, t | $J\ K^{-1}\ m^{-2}$ |
| heat capacity of surface layer | $\gamma_{surf}$ | x, y, t | $J\ K^{-1}\ m^{-2}$ |
| humidity tendency due to precipitation | $\Delta q_{precip}$ | x, y, t | $kg\ kg^{-1}\ s^{-1}$ |

Formatted Table

Formatted Table

| | | | |
|---|---|---|---|
| humidity tendency due to correction | $\Delta q_{correct}$ | x, y, t | $kg\,kg^{-1}\,s^{-1}$ |
| humidity tendency due to evaporation | $\Delta q_{eva}$ | x, y, t | $kg\,kg^{-1}\,s^{-1}$ |
| humidity tendency due to precipitation | $\Delta q_{precip}$ | x, y, t | $kg\,kg^{-1}\,s^{-1}$ |
| ocean temperature tendency due to entertainment | $\Delta To_{entrain}$ | x, y, t | $K$ |
| air diffusion rate | $\kappa_a$ | constant | $4 \times 10^6\,m^2\,s^{-1}$ |
| topography scaling parameter for heat transport | $\rho_a$ | t | $kg\,m^{-3}$ |
| topography correction parameter for water vapor transport | $\rho_v$ | t | $kg\,m^{-3}$ |
| climate mean of air vertical velocity | $\omega_{mean}$ | x, y | $Pa\,s^{-1}$ |
| standard deviation of air vertical velocity | $\omega_{SD}$ | x, y | $Pa^2\,s^{-2}$ |

[Figure]

**Figure 1: The parameterization of the surface albedo in the original GREB (a) and the GREB-ISM (b).**

[Figure]

**Figure 2: Bed rock (unit: m) in the GREB-ISM.**

005

[Figure]

**Figure 3: Temperature response (unit: °C) comparison between NOISM experiment (a, b), FULL experiment (c, d), their difference (e, f; FULL – NOISM, values are plotted on a logarithmic scale) and surface elevation change (g, h) in FULL experiment under CO₂ (left column) reduction and solar radiation reduction (right column) scenario. The temperature response is defined by using equilibrium ice thickness in scenario experiment (100 kyr) minus control experiment (30 kyr for FULL experiment, 50 yrs for NOISM experiment).**

010

[Figure]

**Figure 4:** Zonal mean surface temperature response (a, c; unit: ºC) and surface elevation (b, d) comparison between (dash line) NOISM experiment and (solid line) FULL under $CO_2$ reduction (upper row) and solar reduction (lower row) scenario. The temperature response is defined by using equilibrium ice thickness in scenario experiment (100 kyr) minus control experiment (30 kyr for FULL experiment, 50 yrs for NOISM experiment). The gray line represents 30ºN latitude.

[Figure]

020

**Figure 5: The time evolution of North Hemisphere high latitude (50 °N north) surface temperature response (solid for FULL experiment, dash for NOISM) under $CO_2$ reduction (a) and solar radiation reduction (b) scenario.**

[Figure]

**Figure 6: Ice thickness response (a,b; unit: km) for FULL experiment and feedback effect (c-l) for individual feedback under CO₂ reduction (left column) and solar radiation reduction (right column) scenario. The ice thickness response is defined by using equilibrium ice thickness in scenario experiment (100 kyr) minus control experiment (30 kyr). The feedback effect is defined as the ice thickness response of the FULL experiment minus experiment without the specific feedback. "N-" represents experiment without specific feedback.**

L025

[Figure]

**Figure 7: Zonal mean ice thickness (a,c; unit: km) in scenario run and zonal mean ice thickness [FULL] – [EXP] (b,d; unit: km) for individual feedback under CO₂ reduction (upper row) and solar radiation reduction (lower row) scenario.**

[Figure]

**Fig 8 Same as Fig 6 but for surface temperature (unit: $^{o}$C). Please note we use log scale colors in panels (c-l).**

1030

[Figure]

**Fig 9: Same as Fig 7 but for surface temperature (unit: °C).**

1035

[Figure]

[Figure]

**Fig 10: Feedback strength of global total ice volume (a, b) and global mean surface temperature (c, d) in different experiments under CO₂ reduction (left column) and solar radiation reduction (right column) scenario. The feedback strength is defined by Eq. (3). The albedo feedback strength in total ice volume has been divided by 10 to do better comparison.**

1040

[Figure]

**Fig 11: Surface albedo (a, c; unit: %) and solar radiation (b, d; downward positive, unit: W m⁻²) between FULL and NALBD experiments ([FULL] – [NALBD]).**

[Figure]

[Figure]

 **Figure 12: Snowfall rate changes in winter (left column; DJF-means) and summer (right column; JJA-means) for the FULL experiment after 20 yrs (upper row) and changes within the FULL experiment after 100 kyrs (lower row). Values are in mm d$^{-1}$.**

[Figure]

[Figure]

**Figure 13: Snowfall rates for land points 60 ºN northwards in the control run (black), the FULL experiment after 20 yrs (red) and after 100 kyrs (blue) as function of calendar month. The numbers inside blankets represent the annual mean of snowfall rate (units: mm d⁻¹).**

060

[Figure]

[Figure]

**Fig 14: Development of ice sheets in the first 3 decades of the FULL and NHEAT experiments at a location in northern central Asia (90 ºE, 70 ºN) with seasonally varying ice cover in the control simulation. Latent heat of melting and shortwave radiation difference (a, d) between FULL and NHEAT experiments in the first three decades under CO₂ (upper row) and solar radiation decrease (lower row) scenarios.** Surface temperature and positive degree days (PDD) (b, e), as well as ice thickness (c, f). The x-axis shows the 12 month seasonal cycles after 1, 2 and 3 decades.

065

[Figure]

**Fig 15: (a) Positive degree days (PDD) in the FULL experiment control run. July Surface temperature difference (b, e) and annual mean ice thickness difference (c, e) after 20 yrs between FULL and NHEAT experiments under $CO_2$ (upper row) and solar radiation reduction (lower row) scenarios.**

1085

[Figure]

 **Figure 16: Surface temperature vs surface elevation for all land points (blue for solar radiation reduction scenario and red for CO$_2$ reduction scenario) in FULL experiment in 60 $^o$N northwards in January.**

[Figure]

**Figure 17: Surface elevation change (a,c; unit: m) and surface temperature change (b,d; unit: ºC) in topography lifting experiment using NSNOW – FULL surface elevation as forcing (upper row) and sea level drop experiment using 140 m sea level drop (lower row).**

095

[Figure]

100 **Fig 18: Land-sea mask change (blue represents from ocean to land grid) between FULL and NSLV experiment under $CO_2$ reduction (a) and solar radiation reduction (b) scenarios.**

[Figure]

**Fig 19: (a, c) The time evolution of surface temperature response difference (red; unit: ºC) and specific humidity response difference (blue; unit: g kg⁻¹), as well as (b, d) surface energy flux terms response difference (unit: W m⁻²) at a tropical site (140 ºE, 10 ºS). The response difference of variable is defined as using variable response in FULL experiment minus the NSLV experiment. The gray line represents the time point sea-land transition occurs.**

110

[Figure]

[Figure]

**Figure A1: Annual mean (first column) of surface temperature (first row; unit: °C) and ice thickness (second row; unit: km) of the control run in the FULL experiment in GREB-ISM, along with the difference from observation (second column). The observed surface temperature data used in this study are derived from the climate mean of monthly surface temperature from the ERA-Interim reanalysis dataset spanning the period 1979-2015 (Dee et al., 2011). The observed ice thickness data in Greenland and Antarctica are obtained from the BedMachine dataset (Morlighem et al., 2017, 2020).**

| | | |
|---|---|---|
| **Page 6: [1] Deleted** | **Zhiang Xie** | **7/3/23 3:51:00 PM** |
| **Page 6: [2] Deleted** | **Dietmar Dommenget** | **6/14/23 3:51:00 PM** |
| **Page 6: [3] Deleted** | **Zhiang Xie** | **6/8/23 2:56:00 PM** |
| **Page 24: [4] Deleted** | **Zhiang Xie** | **6/29/23 8:46:00 PM** |
| **Page 24: [5] Deleted** | **Zhiang Xie** | **6/29/23 9:06:00 PM** |
| **Page 24: [6] Deleted** | **Zhiang Xie** | **6/29/23 9:05:00 PM** |
| **Page 25: [7] Deleted** | **Zhiang Xie** | **6/29/23 9:05:00 PM** |
| **Page 25: [8] Deleted** | **Zhiang Xie** | **6/29/23 9:06:00 PM** |
| **Page 25: [9] Deleted** | **Zhiang Xie** | **6/29/23 9:06:00 PM** |
| **Page 26: [10] Deleted** | **Zhiang Xie** | **6/29/23 9:06:00 PM** |
| **Page 26: [11] Deleted** | **Zhiang Xie** | **6/29/23 9:06:00 PM** |